# Experimental realization of temporal refraction and reflection in elastic beams

Shaoyun Wang[1,6], Nan Shao[1,6], Hui Chen [2], Jiaji Chen[1], Honghua Qian[1], Qian Wu[1], Huiling Duan [3], Andrea Alú [4,5] ✉ & Guoliang Huang [3] ✉

Wave reflection and refraction at a time interface follow different conservation laws compared to conventional scattering at a spatial interface. This study presents the experimental demonstration of refraction and reflection of flexural waves across a temporal boundary in a continuum-based mechanical metabeam, and unveils opportunities that emerge by tailoring temporal scattering phenomena for phononic applications. We observe these phenomena in an elastic beam attached to an array of piezoelectric patches that can vary in time the effective elastic properties of the beam. Frequency conversion and phase conjugation are observed upon a single temporal interface. These results are consistent with the temporal Snell's law and Fresnel equations for temporal interfaces. Further, we illustrate the manipulation of amplitude and frequency spectra of flexural wave temporal refraction and reflection through multi-stepped temporal interfaces. Finally, by implementing a smooth time variation of wave impedance, we numerically and experimentally demonstrate the capabilities of the temporal metabeam to realize waveform morphing and information coding. Our findings enable precise control over wave amplitude and frequency through temporally modulated mechanical systems, providing a concrete framework for designing time-mechanical metamaterials and time-phononic crystals.

Wave propagation in time-varying media exhibits dynamics that are strikingly different from those in space-varying media[1–3]. The breaking of time translation symmetry allows energy exchange between the wave and the time-varying medium due to Noether's theorem[4,5], leading to phenomena such as frequency manipulation[6], wave amplification[7], and self-emission[8–10]. Additionally, the Kramers-Kronig relations, grounded in the principle of causality in time-varying systems, enable these systems to surpass well-established fundamental performance limits, including the Bode-Fano limit, Rozanov bound, and Chu limit[11–14].

Despite these differences, wave propagation in time-varying media and space-varying media shares notable analogies. Space-time

duality suggests that phenomena observed in spatially varying systems generally have analogous temporal versions, and vice versa[15–17]. In suddenly varying systems, phenomena analogous to wave scattering at spatial interfaces—such as reflection and refraction[18–23], anti-reflection coatings[24], total internal reflection[25], Goos-Hänchen effect[26,27], wave holography[28], double slit diffraction[29], and photon collisions[30], can occur at temporal interfaces. Moreover, in periodic and disordered systems, temporal analogs such as time crystals[31–33], disordered time-varying media[34–36], k-bandgaps[37], the topology of k-bands and corresponding temporal interface modes[38], and k-gap solitons and breathers[39,40], are developed. In slowly (adiabatically) varying systems,

[1]Department of Mechanical and Aerospace Engineering, University of Missouri, Columbia, MO, USA. [2]Center for Mechanics Plus under Extreme Environments, School of Mechanical Engineering and Mechanics, Ningbo University, Ningbo, China. [3]Department of Mechanics and Engineering Science, College of Engineering, Peking University, Beijing, P R China. [4]Photonics Initiative, Advanced Science Research Center, City University of New York, New York, NY, USA. [5]Physics Program, Graduate Center, City University of New York, New York, NY, USA. [6]These authors contributed equally: Shaoyun Wang, Nan Shao. ✉e-mail: aalu@gc.cuny.edu; guohuang@pku.edu.cn

perfect state transitions such as controllable frequency shifts[41–43], topological state pumping[44,45], and non-Abelian braiding[46,47] are achieved in both space-varying and time-varying systems.

Among these phenomena, wave refraction and reflection at temporal interfaces are considered one of the most fundamental phenomena at the foundations of time crystals, yet they are often challenging to achieve in practical wave systems. One of the key challenges is the creation of a temporal boundary, which typically requires a spatially uniform, ultrafast, and large change of wave impedance[19,48]. The associated experimental challenges keep the experimental study of wave scattering at temporal interfaces in its infancy, particularly in the context of elastic media. Recently, the observation of temporal refraction and reflection has been reported for electromagnetic waves leveraging transmission-line metamaterials[19,21]. Building on this, temporal interfaces are demonstrated in discrete elastic systems using repelling magnets and electromagnetic control[49], but such platforms cannot be extended to continuum systems due to the slow response of electromagnetic actuation. In contrast, prior studies in elastic continua demonstrate smooth temporal stiffness-damping modulation for inducing amplitude modulation and spectral shaping[50], and periodic modulation for $k$-space bandgaps[37] and dynamic phononic crystal[51]. Unlike previous works, which employed stiffness modulation that is continuous, periodic, or slow, realizing a sharp and spatially uniform temporal interface remains elusive in continuum elastic media, primarily due to the strict requirements of sub-microsecond synchronization and uniform stiffness modulation across the entire structure. Although temporal Snell's law and Fresnel equations are well-established in optics and discrete elastic systems, no field-based formulation exists for continuum beams. Likewise, momentum conservation and the breakdown of energy conservation—concepts rooted in Noether's theorem—are not formulated in time-varying beam theory. Furthermore, the use of sharp temporal interfaces for simultaneous and broadband manipulation of multiple wave attributes—frequency, amplitude, and phase—remains largely unexplored in elastic media.

Elastic beams, equipped with piezoelectric patches connected to digital and analog circuits, provide an excellent platform to achieve unconventional elastic wave phenomena, including the non-Hermitian skin effect[52], odd mass density[53], Willis responses[54], frequency conversion[55], and topological pumping[56]. In this study, we report the experimental realization of a sharp temporal interface in an elastic beam, enabled by real-time, circuit-driven stiffness modulation at the sub-microsecond scale using piezoelectric patches. Specifically, we implement sudden transitions between electrical boundary conditions by precisely controlling the switching of shunted circuits in sub-microsecond resolution (0.1 μs). This setup allows the direct observation of temporal refraction and reflection of flexural waves to address a long-standing experimental challenge in the field. We further derive temporal Snell's law and Fresnel equations for flexural waves directly from the governing equations of motion and formulate momentum conservation and the breakdown of energy conservation in time-varying elastic systems using Noether's theorem. These theoretical developments are further validated through numerical simulation and experimental observation, enabled by the ultrafast, broadband, and programmable stiffness modulation. By introducing multiple temporal interfaces, we demonstrate further control over the manipulation of flexural waves in both amplitude and frequency spectra, which have not been explored before. Finally, by programming a smooth time-varying transfer function to realize adiabatic stiffness modulations, we demonstrate additional capabilities in shaping the time-scattered waves in periodic and aperiodic fashion for smart waveform morphing and information coding. These results not only establish a programmable platform for manipulating elastic waves in practical engineering systems but also deepen the fundamental understanding of wave-matter interactions under temporal modulation.

## Results

### A time-varying metabeam supporting temporal interfaces

The time-varying metamaterial under analysis consists of a long, thin beam where bending is the primary mode of deformation. Each unit cell of this metabeam is equipped with a piezoelectric patch that senses bending deformation and provides a self-response. The patch acts as a sensor by generating a voltage proportional to the elongation or contraction of the beam's top surface (see Fig. 1a). The time-dependent transfer function, $H(t)$, comprises an analog switch, a microcontroller, and a time-varying digital potentiometer $R_1(t)$, defined as $H(t) = R_1(t)/$

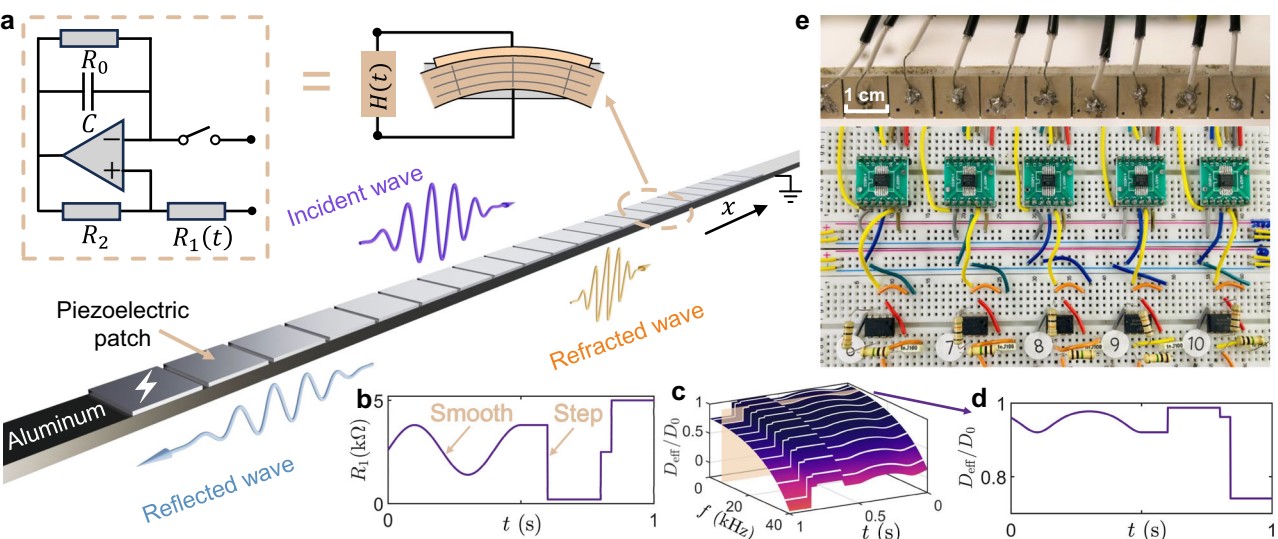

**Fig. 1 | Design of a time-varying metabeam. a** A segment of the time-varying metabeam designed to enable temporal scattering phenomena, such as temporal refraction and reflection, includes a unit cell equipped with a piezoelectric patch and voltage excitation (lighting sign), where the time-varying transfer function is implemented using an electric circuit network. **b** A plot illustrating a time-varying transfer function $H(t)$, which includes both smooth and step changes over time. **c** A graph showing the variation in effective bending stiffness as a function of both frequency $f$ and time $t$, based on the time-varying transfer function $H(t)$ from (**b**). Here, the bending stiffness $D_0$ in open circuit status is 0.88 Nm⁻². **d** The effective bending stiffness $D(t)$ plotted over time along the gray dashed line in (**c**), illustrating its time-dependent behavior. **e** Photo of the metabeam connected with time-varying electronic circuits labeled in each unit cell.

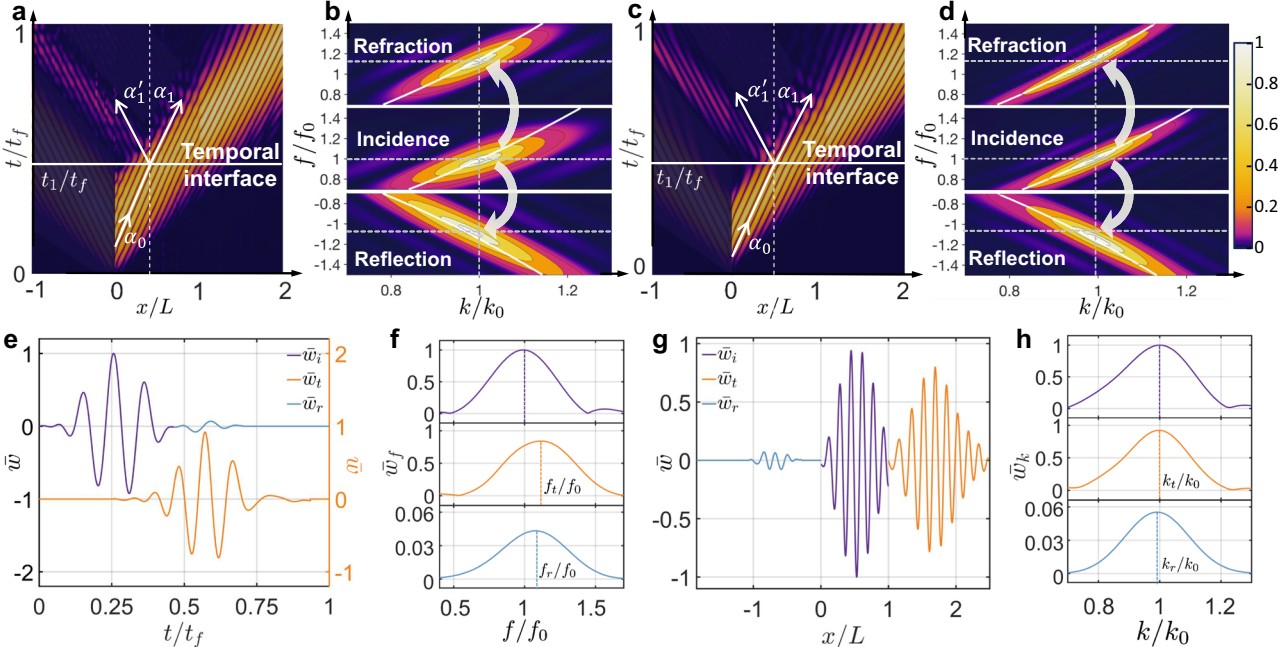

**Fig. 2 | Refraction and reflection of flexural waves at a temporal interface. a**, **c** The space-time diagram of experimental (**a**) and simulated (**c**) wave scattering at a temporal interface $t_1 = 0.44$ ms for an incident wave packet consisting of 3 cycles in the time domain with $t_f = 1.1$ ms. The metabeam is positioned within the interval $(0, L)$, while the surrounding regions consist of an aluminum beam. The angles $\alpha_i$, $\alpha_1$, and $\alpha_1'$ represent the incident, refracted, and reflected angles, respectively. In **a**, these angles are 49°, 52°, and 52.5°, while in **c**, they are 47°, 50. 5°, and 51°. **b**, **d** The top, middle, and bottom panels display the experimental (**b**) and simulated (**d**) contour diagrams of the refracted, incident, and reflected waves, respectively. The data is obtained from a 2D Fourier transformation of the

experimental and simulated data shown in (**a**) and (**c**). The incident frequency $f_0$ is 6.2 kHz and the incident wavenumber $k_0$ is 119 rad/m. The white lines indicate dispersion curves from unit cell analysis. **e** The normalized incident ($\bar{w}_i$) and reflected ($\bar{w}_r$) signals measured at $x/L = 0.05$, along with the refracted signal ($\bar{w}_t$) observed at $x/L = 1$. **f** Spectral analysis of the time-domain signals from (**e**). Here, $f_0$, $f_t$, and $f_r$ are the spectral peak frequencies of incident, refracted, and reflected waves, respectively. **g** The incident spatial profile ($\bar{w}_i$) measured at $t/t_f = 1/3$, along with the refracted ($\bar{w}_t$) and reflected ($\bar{w}_r$) spatial profiles at $t/t_f = 2/3$. **h** Spectral analysis of the spatial-domain signals from (**g**). Here, $k_0$, $k_t$, and $k_r$ are the spectral peak wavenumbers of incident, refracted, and reflected waves, respectively.

$(R_2C)^{57–59}$, as illustrated in Fig. 1b. As a result, the piezoelectric patch also serves as a mechanical actuator, elongating and contracting in response to the applied voltage, and thereby modifying the effective bending stiffness of the beam. To evaluate the performance of the metabeam, we numerically calculated the time-varying effective bending stiffness using COMSOL simulations (see Supplementary Section 1 for details). The resultant effective time-varying bending stiffness of the metabeam $D(t)$ across different frequencies and times can be found in Fig. 1c. Specifically, Fig. 1d illustrates the effective bending stiffness as a function of time for a particular frequency, intersected by the orange surface in Fig. 1c.

We conducted a thorough experimental investigation of the refraction and reflection of flexural waves at a temporal interface. We employ a one-dimensional metabeam controlled by a time-varying electric circuit network, as illustrated in Fig. 1e. The temporal interface is created by toggling the analog switch between ON and OFF states. When the switch is ON, $R_1$ is set to 5 kΩ, corresponding to a bending stiffness of 0.63 N m$^{-2}$. When $R_1$ is OFF, $R_1$ becomes effectively infinite ($\infty$), resulting in a bending stiffness of 0.88 N m$^{-2}$. In the experiment, a 3-cycle tone-burst signal with a central frequency of 6 kHz is applied at the left interface ($x/L = 0$) between the host beam and the modulated metabeam. The magnitude of the flexural wave field, $|w(x, t)|$, is measured throughout the system using a scanning laser Doppler vibrometer (Polytec PSV-400), as shown in Fig. 2a. At $t_1/t_f = 0.4$, the switch transitions from ON to OFF, creating a step-change boundary that causes a rapid shift in the bending stiffness of the modulated metabeam section. The switching time is 150 ns, ensuring an ideal temporal interface. Further details on the experimental setup can be found in the Methods and Supplementary Section 2. At the temporal interface, the incident wave splits into a temporally right-propagating refracted wave and a temporally left-propagating

reflected wave. In this figure, only the temporal reflection is shown, with the left-propagating waves from incidence and spatial reflection at $x/L = 1$ removed (see details in Supplementary Section 3). We define the wavefront direction as the direction of wave propagation (see Fig. 2a, c). As such, the incident angle (49° in the experiment and 47° in the simulation) differs from both the refracted angle (52° in the experiment and 50. 5° in the simulation) and the reflected angle (52. 5° in the experiment and 51° in the simulation), indicating a change in wave direction. The Fourier transform in Fig. 2b illustrates the frequency bandwidth of the input and output waves resulting from temporal reflection and refraction. It shows that the normalized frequency of the incident wave shifts from 1 to 1.16 for refraction and to 1.13 for reflection, while the normalized wavenumbers remain constant. Numerical simulations are performed to validate our experimental observations, as shown in Fig. 2c, d. The results exhibit excellent agreement between the measured output frequencies after the temporal boundary and the numerical predictions, both in the time and frequency domains. To further verify the frequency conversion and wavenumber invariance, time-domain signals measured at $x/L = 0.05$ and $x/L = 1$ are shown in Fig. 2e, where three distinct wave packets corresponding to the incident, refracted, and reflected waves are clearly visible. The normalized frequency $f_t/f_0 = 1.15$ for the refracted wave and $f_r/f_0 = 1.11$ for the reflected wave quantitatively demonstrate the shift relative to the input frequency of the incident wave, as shown in Fig. 2f, indicating a breakdown of energy conservation. Additionally, the spatial-domain signals measured at $t/t_f = 1/3$ and $t/t_f = 2/3$ are shown in Fig. 2g. The wavenumbers $k_t$ for the refracted wave and $k_r$ for the reflected wave are consistent with the wavenumber $k_0$ of the incident wave, as depicted in Fig. 2h, demonstrating the conservation of momentum. The corresponding numerical results for

Fig. 2e–h are provided in Supplementary Section 4. Further results on wave refraction and reflection during the switch from OFF to ON at different frequencies are provided in Supplementary Sections 5 and 6. The effect of finite switching time on wave refraction and reflection is discussed in Supplementary Section 7. To rule out spatial reflection, we simulate an asymmetric pair of wave packets, resulting in a reversed order of the reflected waves (see Supplementary Section 8 for details).

## Refraction and reflection of the flexural wave at a temporal interface

To understand frequency conversion through a temporal interface, we theoretically analyze flexural wave scattering at a temporal interface. In the absence of external forces, the behavior of flexural waves is governed by the Euler-Bernoulli beam equation with time-dependent bending stiffness, expressed as:

$$\frac{\partial}{\partial t}\left(\rho A \frac{\partial w(x,t)}{\partial t}\right) + \frac{\partial^2}{\partial x^2}\left(D(t)\frac{\partial^2 w(x,t)}{\partial x^2}\right) = 0, \tag{1}$$

where $D = EI$ represents the bending stiffness, $E$ is Young's modulus, and $I$ is the second moment of area. Additionally, $\rho$ denotes the density, and $A$ represents the beam's cross-sectional area. For temporal refraction and reflection, the bending stiffness is modulated as a step function over time: $D(t) = D_0 + (D_1 - D_0)\Theta(t - t_1)$, where $\Theta(t)$ is the Heaviside step function, $D_0$ represents the initial bending stiffness, and $D_1$ denotes the bending stiffness after an abrupt change at time $t = t_1$. The conditions for temporal continuity, outlined in Supplementary Section 9, guarantee the smooth transition of both momentum and displacement at the temporal interface without external forces. These conditions are formulated as follows:

$$\rho A \frac{\partial w}{\partial t}\bigg|_{t=t_1^+} = \rho A \frac{\partial w}{\partial t}\bigg|_{t=t_1^-}, \quad w|_{t=t_1^+} = w|_{t=t_1^-}. \tag{2}$$

Here, continuity conditions are applied to momentum and displacement fields, in analogy to the continuity of electric displacement **D** and magnetic flux density **B** in electrodynamics[2]. In our system, the absence of impulses ensures the continuity of momentum. Meanwhile, the invariance of density leads to the continuity of the velocity field, which, in turn, ensures the continuity of the displacement field.

The bending stiffness in Eq. (1) is constant in time at all instances, except at the time interface. Therefore, the wave obeys the conventional expressions stemming from the separation of variables both before and after the time interface. For medium 1 ($t < t_1$), the solution for the incident wave, based on Eq. (1), can be expressed as:

$$w = A_i e^{ik_0 x - i\omega_0 t}, \quad t < t_1, \tag{3}$$

where $A_i$ is the incident wave coefficient, and the angular frequency $\Omega_0$ and wavenumber $k_0$ before the switching event satisfy the dispersion relation $\omega_0 = \sqrt{D_0/(\rho A)}k_0^2$. For medium 2 ($t > t_1$), the displacement field, composed of the refracted and reflected waves after the switching event, can be expressed as:

$$w = \left[Te^{-i\omega_1(t-t_1)} + Re^{i\omega_1(t-t_1)}\right]A_i e^{i(k_1 x - \omega_0 t_1)}, \quad t > t_1, \tag{4}$$

where $T$ is the refraction coefficient, and $R$ is the reflection coefficient. The angular frequency $\Omega_1$ and wavenumber $k_1$ after the switching event are related by the equation $\omega_1 = \sqrt{D_1/(\rho A)}k_1^2$. By inserting the wave solutions from Eqs. (3) and (4) into the temporal continuity conditions given by Eq. (2), we obtain

$$\begin{aligned} e^{ik_0 x} &= (T + R)e^{ik_1 x}, \\ -i\omega_0 e^{ik_0 x} &= (-i\omega_1 T + i\omega_1 R)e^{ik_1 x}. \end{aligned} \tag{5}$$

The temporal continuity conditions in Eq. (5) are satisfied at every point in space, which requires

$$k_1 = k_0, \tag{6}$$

or equivalently

$$\omega_1 n_1 = \omega_0 n_0, \tag{7}$$

where the elastic index of refraction is defined as $n_j = \sqrt{\rho A / D_j}$, with $j = 0, 1$ representing the different media before and after the switching event. Eq. (7) can be interpreted as the temporal Snell's law. Given the frequency of the incident wave and the refractive indices before and after the switching event, the frequency of the refracted wave can be predicted using Eq. (7). The more familiar form, which describes the geometric relationship between the angles of the incident and refracted waves in a space-time diagram, is given by

$$\frac{\tan \alpha_1}{\tan \alpha_0} = \frac{n_0}{n_1}, \tag{8}$$

where $\alpha_0$ is the incident angle and $\alpha_1$ is the refracted angle. See Supplementary Section 10 for the detailed derivation.

By substituting Eq. (6) and Eq. (7) into Eq. (5), we can obtain the temporal scattering coefficients as

$$R = \frac{1}{2}\left(1 - \frac{Z_0}{Z_1}\right), \quad T = \frac{1}{2}\left(1 + \frac{Z_0}{Z_1}\right), \tag{9}$$

where $Z_j = \sqrt{\rho A D_j}$ with $j = 0, 1$ represents the elastic impedance. Eq. (9) serves as the analog of Fresnel equations at the temporal interface, predicting the amplitudes of the refracted and reflected waves. The above derivation is based on the Euler–Bernoulli model, which considers only bending deformation. A detailed justification for neglecting shear deformation (via the Timoshenko beam model), as well as the negligible influence of higher-order flexural modes, longitudinal modes, damping, and nonlinear effects, is provided in Supplementary Section 11.

At a temporal interface, time-translation symmetry is broken, leading to the breakdown of energy conservation according to Noether's theorem. In Supplementary Section 12, the total energy of the elastic beam is given by

$$H = \int dx \left(\frac{1}{2}\rho A \, \partial_t w^\dagger(x,t)\partial_t w(x,t) + \frac{1}{2}EI \, \partial_{xx} w^\dagger(x,t)\partial_{xx} w(x,t)\right), \tag{10}$$

where † denotes the Hermitian conjugate. Before the temporal interface, the energy is

$$H_0 = \frac{1}{Z_0^2}\rho A A_i^2, \tag{11}$$

and after the interface it becomes

$$H_1 = \frac{Z_0^4 + Z_0^2 Z_1^2}{2Z_1^4}H_0, \tag{12}$$

which differs from $H_0$ when $Z_1 \neq Z_0$.

By contrast, the system preserves space-translation invariance, ensuring that momentum is conserved. This conserved momentum,

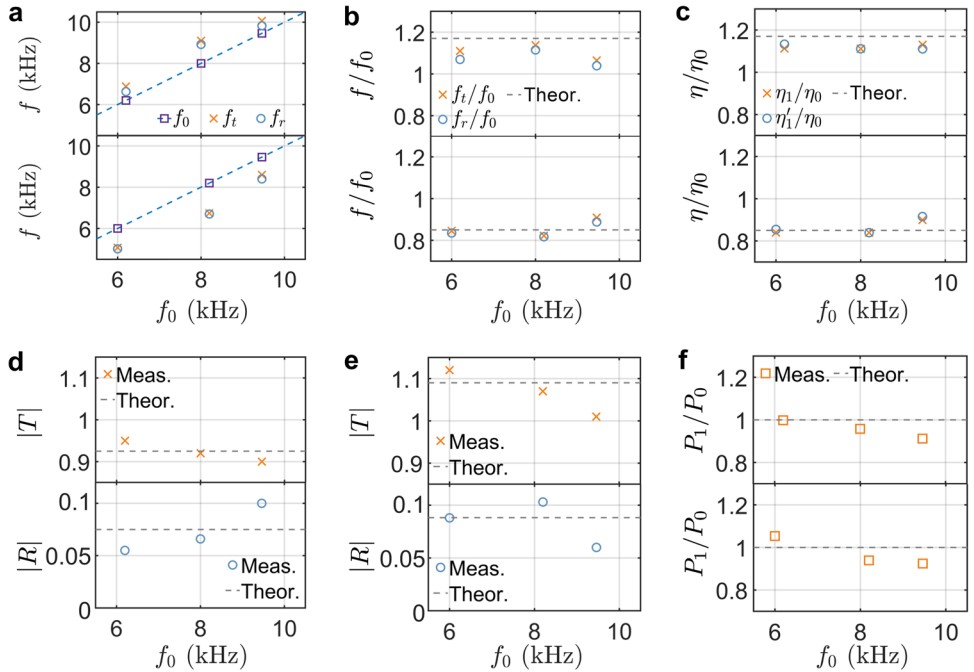

**Fig. 3 | Verification of temporal Snell's law, Fresnel equations, and momentum conservation. a** The measured frequencies of the incident (squares) and reflected (circles) signals at $x/L = 0.05$, and the refracted signal (crosses) observed at $x/L = 1$, plotted against the incident frequency $f_0$. The top (bottom) panel represents the results for the switch from ON to OFF (from OFF to ON). **b** Normalized frequencies of the refracted and reflected waves, along with the theoretical prediction of Snell's law (dashed line). The top (bottom) panel represents the results for the switch from ON to OFF (from OFF to ON). **c** The measured ratio of the tangent of the refraction angle ($\eta_1 = \tan \alpha_1$) to the tangent of the incidence angle ($\eta_0 = \tan \alpha_0$), along with the measured ratio of the tangent of the reflection angle ($\eta_1' = \tan \alpha_1'$) to the tangent of the incidence angle, is compared with the theoretical prediction based on Snell's

law (represented by the dashed line). The top (bottom) panel represents the results for the switch from ON to OFF (from OFF to ON). **d** The measured and theoretical magnitudes of the refraction coefficient (top panel) and reflection coefficient (bottom panel) at $t/t_f = 2/3$ for the switch from ON to OFF, plotted against the incident frequency $f_0$. **e** The measured and theoretical magnitudes of the refraction coefficient (top panel) and reflection coefficient (bottom panel) at $t/t_f = 2/3$ for the switch from OFF to ON, plotted against the incident frequency $f_0$. **f** The ratio of momentum before and after the temporal interface, along with the theoretical prediction based on momentum conservation (shown as the dashed line). The top panel shows the results for the switch from ON to OFF, while the bottom panel corresponds to the switch from OFF to ON.

also known as Noether's charge, of the elastic beam is given by

$$P = \int \rho A \left[ (\partial_t w)^\dagger \partial_x w + \partial_t w (\partial_x w)^\dagger \right] dx, \tag{13}$$

where † denotes the Hermitian conjugate. The detailed derivation of Eq. (13) using the complex scalar field theory of the Euler-Bernoulli beam is provided in Supplementary Section 12. The momentum of the wave before the time switching is

$$P_0 = 2\rho A \omega_0 k_0 A_i^2, \tag{14}$$

whereas the momentum of the waves after the switching time is

$$P_1 = 2\rho A \omega_1 k_0 (T^2 - R^2) A_i^2. \tag{15}$$

With the aid of Eq. (9), the conservation of momentum can be easily verified as

$$P_0 = P_1 = 2Z_0 k_0^3 A_i^2. \tag{16}$$

In addition, the momentum of both the incident and scattered waves is proportional to $k_0^3$, indicating that the wavenumber remains invariant. It is worth noting that although the full electromechanical system conserves energy, our analysis treats the mechanical beam as an open subsystem. The shunting circuit serves as an engineered environment that modulates stiffness, resulting in mechanical energy exchange and thus energy non-conservation, while momentum remains conserved through spatial translation symmetry.

Guided by the derived Snell's law in Eq. (7), the frequency conversion capability for different incident frequencies at the temporal interface is further tested experimentally. Figure 3a presents the measured central frequencies of the refracted (crosses) and reflected (circles) waves at the temporal interface as a function of the incident frequency $f_0$ (squares), for both the switch from ON to OFF (top panel) and OFF to ON (bottom panel). In the figure, the corresponding time-domain signals are measured at $x/L = 0.05$ for the incident and reflected waves, and at $x/L = 1$ for the refracted wave. The frequencies of the refracted and reflected waves shift upward (downward) during the transition from ON to OFF (OFF to ON), confirming the occurrence of frequency conversion. Furthermore, the frequencies of the refracted and reflected waves, normalized by the frequency of the incident wave, are presented in Fig. 3b. Additionally, Fig. 3c presents the tangents of the angles for the incident, refracted, and reflected waves, with the corresponding angles provided in Supplementary Table 3. The discrepancy observed for the switch from ON to OFF at 6 kHz arises because low-frequency incident signals do not terminate before reaching the temporal boundary, while the discrepancy at high frequencies is attributed to the limitations of the homogeneous beam model at short wavelengths. Therefore, the 6–10 kHz range is deliberately selected to balance the wavelength considerations and maintain the validity of the continuum beam model. Overall, these normalized frequencies and tangent ratios agree with the theoretical predictions ($n_0/n_1 = 1.17$ in the top panel and $n_0/n_1 = 0.85$ in the bottom panel) derived from Snell's law in Eq. (7) and Eq. (8). This agreement confirms that the observed frequency conversion and directional changes are consistent with the analytical predictions based on the temporal analog of Snell's law.

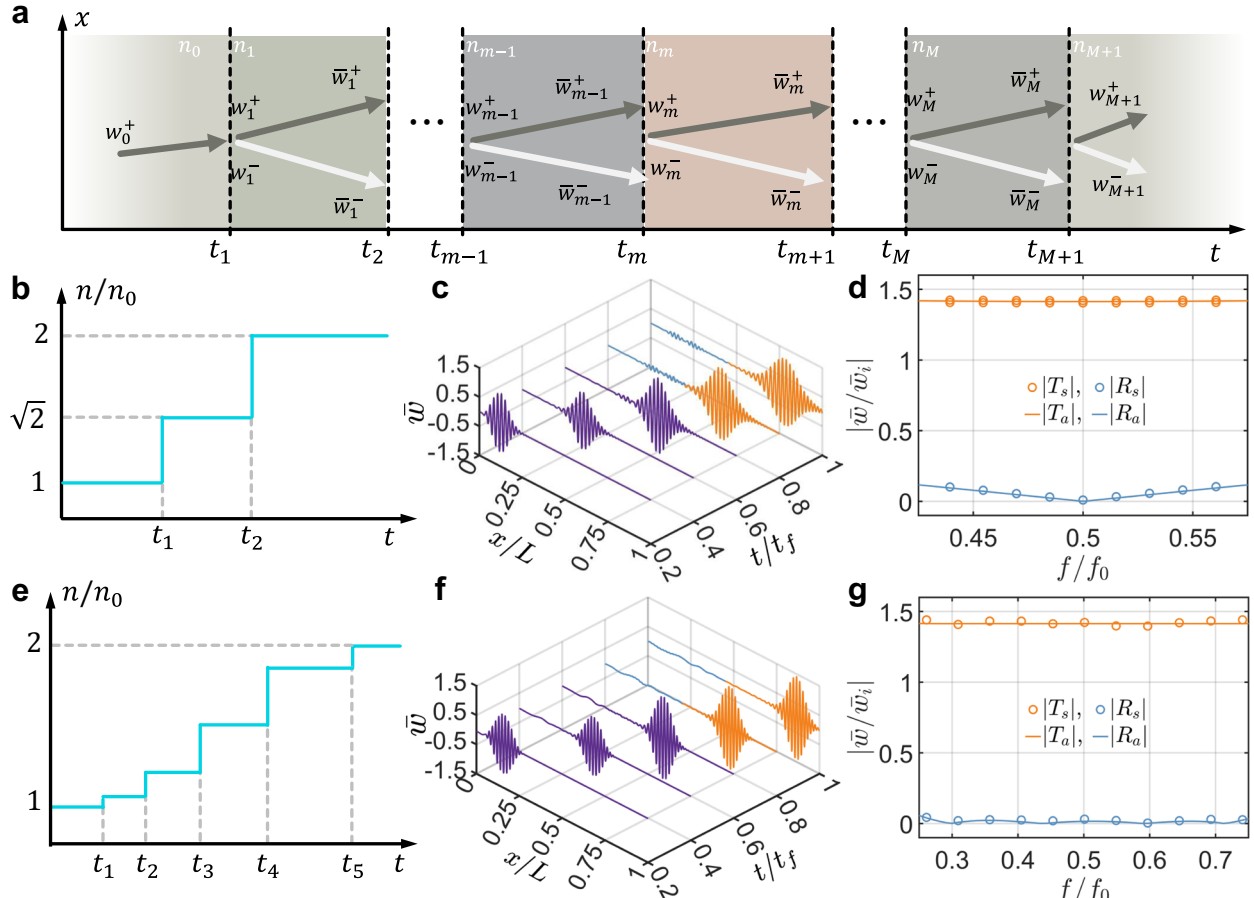

**Fig. 4 | Multi-stepped temporal interfaces for anti-reflection. a** Schematic diagram illustrating a wave propagating through multi-stepped temporal interfaces. The setup consists of two temporally semi-infinite media with a cascade of $M$ temporal slabs, separated by $M + 1$ temporal interfaces. **b, e** The optimal time-dependent index of refraction function designed for anti-reflection of a single frequency (**b**) and broadband frequency (**e**). **c, f** Wave packet evolution by COMSOL simulation for verifying anti-reflection of a single frequency (**c**) and broadband frequency (**f**). The wave packet, with a central frequency of 6 kHz, consists of 5 cycles in time, with $t_f = 6$ ms and $L = 2.56$ m. **d, g** Refraction and reflection coefficients as a function of the normalized incident frequency $f/f_0$ for single frequency (**d**) and broadband frequency (**g**) anti-elimination, where $f_0 = 6$ kHz, with circles representing numerical simulation results and solid lines representing analytical calculations using the transfer matrix method. The subscripts "s" and "a" denote simulated and analytical results, respectively. The results are obtained from the Fourier transform at $t/t_f = 0.75$ in (**c**) and (**f**).

The derived Fresnel equations in Eq. (9) across the temporal boundary are also verified. The amplitudes of the incident, refracted, and reflected waves are defined as the height of spectral peaks, as shown in Fig. 2h. The corresponding spatial-domain signals are measured at $t/t_f = 1/3$ for the incident wave, and at $t/t_f = 2/3$ for the refracted and reflected waves. Figure 3d, e show the measured amplitudes of the refraction coefficient ($T$, crosses), reflection coefficient ($R$, circles), and their theoretical predictions (dashed lines) based on Eq. (9), as functions of the incident frequency for the switches from ON to OFF and OFF to ON, respectively. In Fig. 3d, the measured amplitudes closely match the theoretical predictions of $|T| = 0.925$ and $|R| = 0.075$ for the switch from ON to OFF at the given incident frequencies. Similarly, the minor deviation between the experimental measurement and theoretical predictions may be attributed to the absence of the low-frequency incident signals and the limitations of the homogeneous beam model. Also, as shown in Fig. 3e, the measured amplitudes are mainly consistent with the theoretical predictions of $|T| = 1.09$ and $|R| = 0.089$ for switching OFF to ON at different incident frequencies. A small deviation is also observed. However, the reasonable consistency verifies the elastic analog of the Fresnel equations across various incident frequencies. Finally, the ratio of momentum before and after the temporal interface is presented in Fig. 3f. The momentum ratios are mainly close to 1 for both the switch from ON to OFF and from OFF to ON, validating momentum conservation.

## Flexural wave engineering with temporal multi-stepped interfaces

Leveraging scattering phenomena at a single temporal interface, we explore the engineering of flexural wave interferences in time, inspired by their photonic counterparts, such as in discrete temporal crystals and temporal metabeams analogous to the optical counterpart[24,60,61]. To demonstrate this, we consider flexural wave propagation through a temporal modulation of the bending stiffness characterized by $M + 1$ stepped temporal interfaces, as depicted in Fig. 4a. The unbounded medium initially has a refractive index $n_0$ for times $t < t_1$. At $t = t_1$, a temporal modulation of the refractive index occurs, characterized by $n_m$, $m = 1, 2, \ldots, M$ over $M$ intervals. This assumption of step temporal transitions is highly idealized, as it implies an infinitely fast response of the medium to the modulation. The theory of wave propagation through multi-stepped temporal interfaces in an unbounded medium is developed using the rigorous transfer matrix approach. The wave components on the left and right sides of the $m$th temporal slab are represented by $[w_m^+, w_m^-]^{\mathrm{T}}$ and $[\tilde{w}_m^+, \tilde{w}_m^-]^{\mathrm{T}}$, respectively. These components are connected by the following expression:

$$\begin{bmatrix} \tilde{w}_m^+ \\ \tilde{w}_m^- \end{bmatrix} = \mathbf{N}_a^m \begin{bmatrix} w_m^+ \\ w_m^- \end{bmatrix}, \quad (17)$$

where the propagation matrix $\mathbf{N}_a^m$ is

$$\mathbf{N}_a^m = \begin{bmatrix} e^{-i\omega_m \Delta t_m} & 0 \\ 0 & e^{i\omega_m \Delta t_m} \end{bmatrix}, \tag{18}$$

and the $m$th time interval is $\Delta t_m = t_{m+1} - t_m$. In addition, due to the continuity equations in Eq. (2), the wave components $[w_m^+, w_m^-]^{\mathrm{T}}$ and $[\tilde{w}_{m-1}^+, \tilde{w}_{m-1}^-]^{\mathrm{T}}$ on either side of $m$th temporal interface satisfy the following relation:

$$\begin{bmatrix} w_m^+ \\ w_m^- \end{bmatrix} = \mathbf{N}_b^m \begin{bmatrix} \tilde{w}_{m-1}^+ \\ \tilde{w}_{m-1}^- \end{bmatrix}. \tag{19}$$

Here, the matching matrix $\mathbf{N}_b^m$ is

$$\mathbf{N}_b^m = \begin{bmatrix} T_m & R_m \\ R_m & T_m \end{bmatrix}. \tag{20}$$

where $T_m = 1 + Z_{m-1}/Z_m$ and $R_m = 1 - Z_{m-1}/Z_m$. Applying Eqs. (17) and (19) recurrently, we obtain the relationship between the waves in the initial and final temporal boundaries:

$$\begin{bmatrix} w_{M+1}^+ \\ w_{M+1}^- \end{bmatrix} = \mathbf{N} \begin{bmatrix} w_0^+ \\ 0 \end{bmatrix}, \tag{21}$$

where the overall transfer matrix $\mathbf{N}$ is given as follows:

$$\mathbf{N}(\boldsymbol{n}, \boldsymbol{t}, f_0) = \mathbf{N}_b^{M+1} \prod_{m=1}^{M} \mathbf{N}_a^m \mathbf{N}_b^m. \tag{22}$$

Here, $\boldsymbol{n}$ and $\boldsymbol{t}$ are vectors representing the refractive indices $n_m$ (for $m = 1, 2, \ldots, M$) and the times at different interfaces $t_m$ (for $m = 1, 2, \ldots, M+1$), respectively. $f_0$ denotes the wave frequency in the left unbounded medium. Additionally, the refraction coefficient is defined as $T(\boldsymbol{n}, \boldsymbol{t}, f_0) = w_{M+1}^+/w_0^+ = N_{11}$, and the reflection coefficient is defined as $R(\boldsymbol{n}, \boldsymbol{t}, f_0) = w_{M+1}^-/w_0^+ = N_{21}$.

Using the analytical solutions for the reflection and refraction coefficients, we will explore three examples of engineered flexural wave propagation across multi-stepped temporal interfaces through inverse design. These examples will leverage temporal intervals and modulated slab parameters, demonstrating the potential of temporal multi-stepped interfaces as versatile wave transformers. First, we investigate anti-reflection temporal coatings by introducing two-stepped temporal slabs with equal travel times to achieve impedance matching and frequency conversion between two connected waveguides with different stiffnesses, analogous to quarter-wavelength impedance matching in the spatial domain[62]. Second, we employ multi-stepped temporal interfaces composed of five temporal slabs to achieve broadband wave anti-reflection[60]. Finally, we propose temporal multi-stepped interfaces with alternating high and low refractive indices to enable wave amplification in both reflection and refraction[61]. The temporal parameters of these multi-stepped structures are determined using an optimization method, which seeks to identify the optimal temporal interface parameters by minimizing a target function related to the reflection and refraction coefficients, subject to specified constraints. Detailed formulations of the optimization problem for the three cases are provided in Supplementary Section 13, and the optimized results will be validated against the analytical solution from Eq. (22) for multi-stepped interfaces using the optimized parameters.

Figure 4b presents the numerically derived two-stepped temporal configuration, consisting of a single temporal slab with a refractive index $n_1$, designed to eliminate the reflection of an incident flexural wave at a frequency of $f_0 = 6$ kHz, closely matching our experimental testing conditions. The duration of the temporal slab is defined as $\Delta t_1 = t_2 - t_1$. The wave initially propagates through a medium with a refractive index of $n_0$, while the final medium has a refractive index of

$n_2 = 2n_0$. By minimizing the square of the reflection coefficient magnitude at the frequency (see Supplementary Section 13 for details), we obtain the optimal values of $n_1 = 1.414n_0$ and $\Delta t_1 = 0.3536/f_0$. These results are in excellent agreement with the analytical solution in Eq. (21), where $n_1 = \sqrt{2}n_0$ and $t_2 = 1/(2\sqrt{2}f_0)$ (see Methods)[24]. Since the time duration $t_2$ equals a quarter of the wave period in the slab, this temporal medium is referred to as a quarter-wave transformer. Physically, as the incident wave passes through two temporal interfaces, it generates two refracted and two reflected waves. In the quarter-wave transformer, the two reflected waves cancel each other through coherent subtraction. Using the optimized parameters, Fig. 4c illustrates the evolution of a wave packet with a central frequency of $f_0 = 6$ kHz as it propagates through the temporal quarter-wave transformer. The simulation is performed in a long temporal metabeam with 240 unit cells (see Supplementary Section 4 for details on refraction and reflection without the temporal slab). In Fig. 4c, the reflected wave packet is almost entirely suppressed, though two small wave packets with different frequencies are generated. This occurs because the transformer perfectly eliminates reflection at the frequency, while residual components remain at other frequencies[24]. Additional simulations were performed for wave packets with different central frequencies, and the corresponding refraction coefficient $T_s$ and reflection coefficient $R_s$ are plotted in Fig. 4d. In this figure, the numerical refraction coefficient $T_s$ and reflection coefficient $R_s$ (circles) closely match the analytical refraction coefficient $T_a$ and reflection coefficient $R_a$ (solid lines), calculated using the transfer matrix method. The reflection coefficient approaches zero near $0.5f_0$ but diverges significantly at other frequencies, confirming that the temporal quarter-wave transformer eliminates reflection optimally and shifts the frequency component to the frequency $0.5f_0$. The magnitude of the frequency shift is determined by the refractive index change through the temporal interfaces. It is also observed that the values of scattering coefficient $T_s$ are greater than unity for most frequencies, revealing a gain effect induced on the propagating signal by the medium, where energy conservation is violated.

To achieve broadband wave anti-reflection, the multi-stepped interfaces are determined using an optimization method. Figure 4e shows the resulting temporal medium with four slabs, characterized by $\boldsymbol{n} = [1.075, 1.271, 1.573, 1.860]n_0$ and $\boldsymbol{t} = [0, 0.262, 0.575, 0.962, 1.416]/f_0$, designed for broadband anti-reflection across the frequency range from $0.5f_0$ to $1.5f_0$. The initial and final refractive indices are $n_0$ and $n_2 = 2n_0$, respectively. These parameters are derived by minimizing the integral of the squared reflection coefficient over the same frequency range (see Supplementary Section 13 for details). The wave packet evolution in this medium is shown in Fig. 4f, where reflections are significantly suppressed, except for minor low- and high-frequency noise. Figure 4g shows the simulated and analytical refraction and reflection coefficients in the frequency domain, with the simulated reflection coefficient staying near zero across the range from $0.25f_0$ to $0.75f_0$, where the frequency shifts relative to the incident frequency. This result is consistent with the analytical results from the transfer matrix method, confirming the accuracy of the optimization method and the effectiveness of temporal multi-stepped interfaces for broadband anti-reflection.

Different from spatial interfaces, time interfaces can provide energy to the input waves; hence, through interference, it is possible to design time scattering profiles that achieve broadband wave amplification in both reflected and refracted waves. Figure 5a presents optimal parameters to achieve this task for a temporal multilayer with three slabs, where $\boldsymbol{n} = [3, 1, 3]n_0$ represents the refractive indices and $\boldsymbol{t} = [0, 0.75, 0.25, 0.75]/f_0$ represents the corresponding time intervals. The initial refractive index is denoted by $n_0$. These parameters are obtained by minimizing the negative square of the magnitude of the refraction coefficient at 6 kHz (see Supplementary Section 13 for details). The length of each slab is a quarter of the wave period,

resulting in a phase decrease of $\pi/4$ for the refracted wave and a phase increase of $\pi/4$ for the reflected wave. After passing through each slab, the phase difference between the refracted and reflected waves reaches $\pi$, leading to constructive interference at the subsequent temporal interface, which amplifies the waves[30]. The wave packet evolution in this medium is shown in Fig. 5b, where significant amplification of both refraction and reflection is observed. Figure 5c shows that the simulated reflection coefficient peaks at ~4.5 around the frequency $f_0$, closely matching the analytical results from the transfer matrix method. Although the refraction coefficient is optimized for a specific frequency, amplification is observed over a broad frequency range, spanning from $0.6f_0$ to $1.4f_0$.

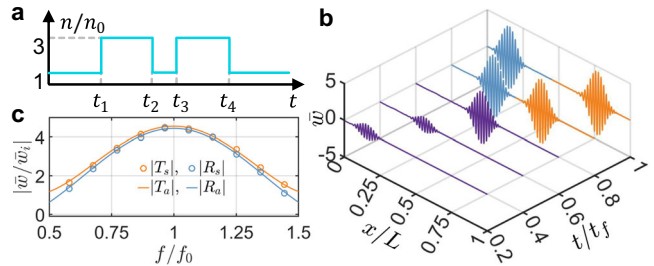

**Fig. 5 | Multi-stepped temporal interfaces for wave amplification. a** The optimal distribution of the index of refraction with 4 temporal interfaces for wave amplification. **b** The evolution of the wave packet is simulated using COMSOL to verify wave amplification. This wave packet, with a central frequency of 6 kHz, consists of 5 cycles for $t_f = 6$ ms and spans a length of $L = 2.56$ m. **c** Refraction and reflection coefficients as a function of the normalized incident frequency $f/f_0$, where $f_0 = 6$ kHz, with circles representing numerical simulation results and solid lines representing analytical calculations using the transfer matrix method. The subscripts "s" and "a" denote simulated and analytical results, respectively. The results are obtained from the Fourier transform of data at $t/t_f = 0.75$ in (**b**).

## Temporal metabeams for smart waveform morphing and information coding

In this section, we propose a method for designing temporal metabeams with tunable time-varying bending stiffness by utilizing self-reconfigurable transfer functions controlled by time-varying digital potentiometers. By programming the time-domain behavior of these digital potentiometers, the metabeam's bending stiffness can be modulated to follow desired periodic or aperiodic patterns. The proposed time-varying metabeam demonstrates capabilities in shaping the amplitudes of transmitted flexural waves in the time domain, both experimentally and numerically, as illustrated in Fig. 6a. It is essential to emphasize that the time-varying parameters must meet adiabatic conditions, meaning the bending stiffness should change gradually enough to avoid inducing frequency conversion at the temporal stepped interface.

Under the assumption of the length of the unit cell being much shorter than the wavelength, the effective bending stiffness of the metabeams could be positive and negative. Previous studies employed metabeams with time-varying negative bending stiffness to modulate the amplitude of flexural waves within the subwavelength Bragg bandgap[59,63]. For the metabeam with the negative stiffness, the flexural wave will exponentially decay in a factor proportional to the magnitude of the effective bending stiffness (see Supplementary Section 1). Therefore, the negative stiffness of the metabeam with temporal variability spanning a reasonable range could result in a significant change in wave transmission. The time-varying bending stiffnesses are difficult to measure directly; hence, we indirectly verify their existence by studying the wave transmission properties of the metabeam. To understand the underlying mechanism of this method, we analyze the influences of the constitutive parameters on the wave transmission of a metabeam with 30 unit cells.

A time-varying transfer function is applied to the metabeam to achieve the desired waveform morphing. For instance, the transfer function can be modulated as a sinusoidal or smooth step function

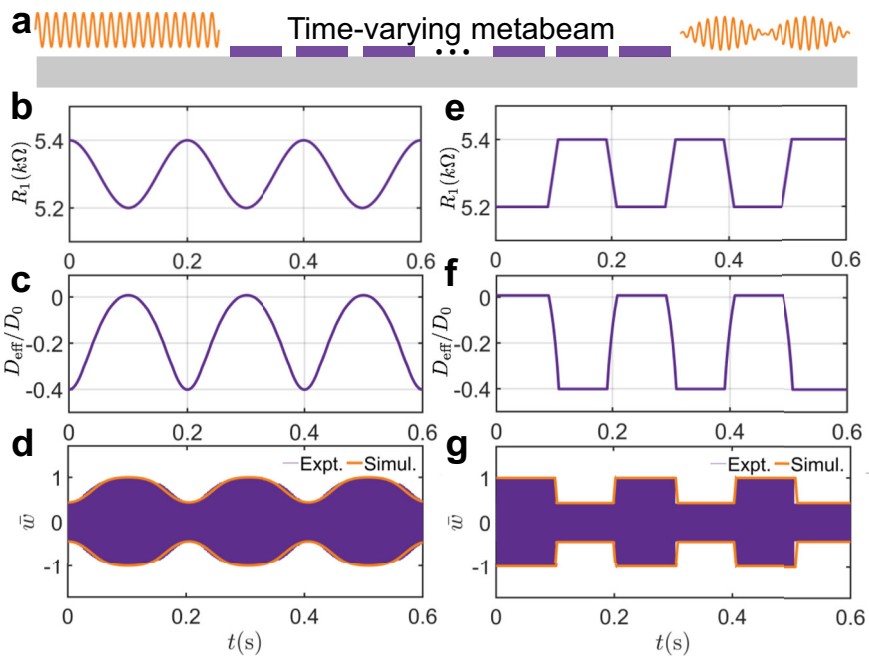

**Fig. 6 | Smart waveform morphing schematic enabled by a smooth time-varying metabeam. a** Schematic illustration of smart waveform morphing enabled by a time-varying metabeam. **b, e** The transfer function can be modulated into various forms, such as a sinusoidal function (**b**) and smooth step function (**e**). **c, f** The corresponding effective bending stiffness over time at frequencies of 33 kHz of sinusoidal function and smooth step function, respectively. **d, g** The simulated response (encapsulated by an orange envelope) and the measured time response (purple lines) at a point on the right side of the beam of the sinusoidal function and smooth step function, respectively. The excitation source, with a frequency of 33 kHz, is located at the leftmost piezoelectric patch.

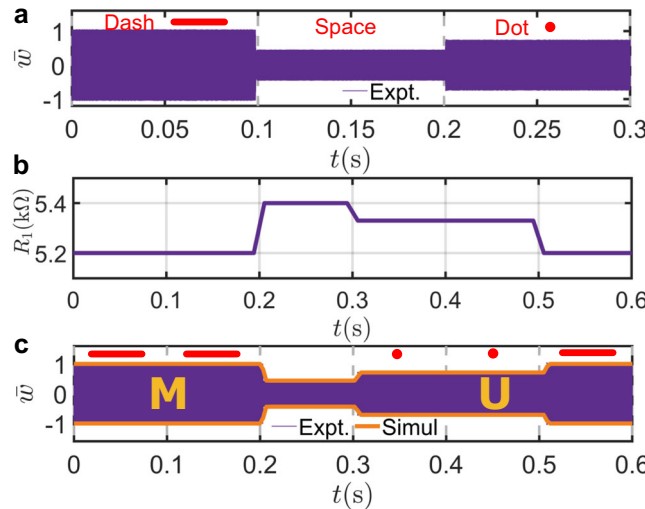

**Fig. 7 | Time-varying metabeam for Morse coding. a** Experimental demonstration of the fundamental Morse code elements—dash, space, and do—using flexural waves with distinct amplitude levels. The experimental variation remains within ±2.5%; the amplitude difference between a dash and a dot exceeds 30%, while that between a space and a dot is -15%. **b** Time profile of the resistance $R_1(t)$, used to modulate the beam's effective stiffness to generate these signals. **c** Comparison of experimental (purple) and simulated (orange) signals encoding the text "MU".

using a time-varying digital potentiometer, as shown in Fig. 6b, e. For an excitation frequency of 33 kHz, the corresponding effective bending stiffness over time is plotted in Fig. 6c, f. An excitation frequency of 33 kHz is chosen because it corresponds to the transition of the effective bending stiffness from positive to negative values under circuit tuning. The detailed smooth functions can be found in Supplementary Section 14. The results in Fig. 6c, f show that the modulation pattern of the bending stiffness can be flexibly tuned to any desired shape by programming the transfer functions. Next, we demonstrate that modulating the constitutive parameters can be used for waveform morphing. Using sinusoidal and smooth step function patterns as examples, we measure the transmitted waves for different modulation amplitudes, as shown in Fig. 6d, g. As shown in Fig. 6d, g, the variations in bending stiffness are clearly reflected in the changing amplitudes of the transmitted wave, demonstrating that the metabeam can modulate the amplitude of transmitted signals in a desired periodic manner. The experimental and simulated transmitted wave amplitudes are in excellent agreement, following the patterns predicted by the homogenized model. This confirms that the adiabatic assumption is satisfied, allowing the metabeam to be treated as a temporal Cauchy-elastic medium with strongly time-modulated constitutive parameters.

As an additional example, we utilize the metabeam as an elastic Morse coder to demonstrate its capability to modulate wave amplitudes in an aperiodic manner. Morse codes represent letters of the alphabet, numerals, and punctuation marks by arranging dots, dashes, and spaces, as depicted in Fig. 7a. Traditionally, the codes are transmitted as electric pulse, mechanical, or visual signals. Here, we program the codes with time-varying bending stiffness of the metabeam. These codes are distinguished by the amplitudes of the transmitted waves when the metabeam is excited on the left side by a constant sine signal at 33 kHz. Specifically, a "dash" is represented by the measured transmitted wave with the largest amplitude among the three states, and it is normalized to unit 1, as shown in Fig. 7a; a dot is represented by the normalized amplitude of the transmitted wave being 0.75; a space between letters is represented by the amplitude of the transmitted wave being 0.45. In addition, each state is designed to last 0.1 seconds in the time domain.

Using the coding rules, we encode "MU", the abbreviation for the University of Missouri, into the metabeam via a time-varying transfer function. The time-varying digital potentiometer corresponding to this transfer function is shown in Fig. 7b. The encoded information can be extracted by mechanically stimulating the left side of the metabeam with a constant 33 kHz sine signal and measuring the transmitted wave on the right side. The measured signal, shown in Fig. 7c, successfully transmits the letters "MU". The measured signals are in excellent agreement with our numerical simulations.

## Discussion

This study presents the experimental observation of temporal refraction and reflection of flexural waves in a time-varying metabeam with time-modulated bending stiffness. We also demonstrated stepped and smooth variations of multiple time interfaces for enhanced wave control based on time scattering. Our metabeam, composed of an elastic beam with attached piezoelectric patches, is a mechanical platform well-suited to explore a wide range of time-varying media in ultrafast wave control and advanced signal processing technologies through time-varying transfer functions. The temporal variation patterns can be of periodic or non-periodic form, allowing for fast and flexible adjustment, and can even be controlled wirelessly. We have established analogs to Snell's law and Fresnel equations for elastic waves, providing a theoretical framework for understanding wave scattering at temporal boundaries, which was validated through experimental testing. Due to spatial translational symmetry, we show that momentum remains conserved, revealing the fundamental principles governing wave scattering in time-varying media. The rapid modulation of stiffness via elastic elements is scalable, and it can be applied to more complex designs, including damping compensation. A smooth time-varying metabeam was then implemented in waveguides, achieving anti-reflection temporal coatings, wave morphing, impedance matching, and efficient frequency conversion. Beyond these remarkable functionalities, the metabeam may serve as a platform to study phenomena like temporal pumping and $k$-space band gaps, inspiring the development of wave-based devices for signal processing.

Moving forward, this work opens exciting opportunities, in particular in the context of combining spatial and temporal interfaces for 4D elastic metamaterials[1]. Opportunities to realize elastic time crystals and quasi-crystals emerge in this platform, enabling the time analogs of sophisticated spatial wave features such as Hofstadter's butterfly and topological modes. Time-varying elastodynamic media introduce additional degrees of freedom for controlling and manipulating wave and material phases, expanding the design space for dynamic wave-based systems. The elastic wave platform introduced here offers interesting opportunities to explore these phenomena experimentally.

## Methods
### Sample fabrication
The metabeam is composed of 30 piezoelectric patches (APC 850: 10 mm × 10 mm × 0.8 mm) mounted via conductive epoxy onto the middle of an aluminum host beam (180 mm × 10 mm × 2 mm). In the circuit, the resistors $R_0 = 1\,M\Omega$ and $R_2 = 10\,k\Omega$ with 5% error, the capacitors are film capacitors with 5% error, the microcontroller is an STM32 Nucleo Development Board with an STM32F446RE MCU, the digital potentiometer $R_1$ is a 20 kΩ AD5291 from Analog Devices, and the analog switch is a DG411 from Vishay Siliconix.

### Experimental procedures
In experiments, 30 piezoelectric patches are connected with control circuits, and another piezoelectric patch on their left is used to generate incident flexural waves. We employ three cycles of tone-burst signals with central frequencies at 6 kHz, 8 kHz, and 10 kHz for

temporal refraction and reflection. We generate and amplify incident wave signals via an arbitrary waveform generator (Tektronix AFG3022C) and a high-voltage amplifier (Krohn-Hite), respectively. Transverse velocity wavefields are measured on the surface of the metamaterial by a scanning laser Doppler vibrometer (Polytec PSV-400). The analog switch is controlled by the microcontroller to turn off 0.3 ms after the reference signal, generated by the arbitrary waveform generator with an amplitude of 3 V, drops below $-2V$. For smart waveform morphing, the excitation is applied using a sinusoidal signal with a frequency of 33 kHz, and the velocity signal is measured at a position 0.2 m to the right of the rightmost piezoelectric patches.

### Finite element simulations

The numerical simulations are conducted by using a 2D "Piezoelectricity, Solid" module in the commercial finite element software COMSOL Multiphysics. The material of the host beam is implemented by Aluminum [solid], and the material of the piezoelectric patches is PZT-5A from the COMSOL Material Library. The current passing through the top surface of the piezoelectric patch is coupled with voltage on the top surface by a transfer function using "Global ODEs and DAEs" module for simulating the circuit effect. The dispersion curves in Fig. 2h are obtained by using eigenfrequency analysis with Floquet boundary conditions. The time-domain analysis is conducted for the same setup as the experiment, where the transfer function is defined as a time-varying function. The simulations are performed using the generalized-$\alpha$ method, with a time step set to 1/100 of the wave period to accurately resolve the dynamic response. The boundary conditions on both sides for temporal refraction and reflection are free boundaries. However, two gradient-damping beams are attached on both sides to create perfect absorption boundary conditions for smart waveform morphing.

### Temporal quarter-wave transformer

The temporal quarter-wave transformer comprises a slab with refractive index $n_1$ surrounded by two semi-infinite media with refractive indices $n_0$ and $n_2$, respectively. Therefore, the transfer matrix consists of two matching matrices and one propagation matrix combined as

$$\mathbf{N} = \begin{bmatrix} T_2 & R_2 \\ R_2 & T_2 \end{bmatrix} \begin{bmatrix} e^{-i\omega_1 \Delta t_1} & 0 \\ 0 & e^{i\omega_1 \Delta t_1} \end{bmatrix} \begin{bmatrix} T_1 & R_1 \\ R_1 & T_1 \end{bmatrix}. \tag{23}$$

Eq. (23) gives the reflection coefficient:

$$R = N_{21} = R_1 T_2 e^{i\omega_1 \Delta t_1} + R_2 T_1 e^{-i\omega_1 \Delta t_1}. \tag{24}$$

The reflection coefficient can be canceled out through coherent subtraction, which implies

$$R_1 T_2 e^{i\omega_1 \Delta t_1} + R_2 T_1 e^{-i\omega_1 \Delta t_1} = 0. \tag{25}$$

This complex equation can be decomposed into two real equations corresponding to the amplitude and the phase:

$$\begin{aligned} \omega_1 \Delta t_1 &= \frac{(2n+1)\pi}{2}, \\ R_2 T_1 &= T_2 R_1. \end{aligned} \tag{26}$$

These two equations give the anti-reflection conditions:

$$\begin{aligned} \Delta t_1 &= \frac{(2p+1)}{4f_1}, \\ n_1 &= \sqrt{n_0 n_2}, \end{aligned} \tag{27}$$

where $p$ is an integer, and $f_1$ is the wave frequency in the middle slab with refractive index $n_1$. If $p = 0$, the slab length $\Delta t_1$ equals a quarter of

the wave period, making this temporal medium a quarter-wave transformer. In the main text, with $n_2 = 2n_0$, we have $n_1 = \sqrt{2}n_0$, $f_1 = \sqrt{2}f_0/2$, and $t_2 = 1/(2\sqrt{2}f_0)$ for $p = 0$ and $t_1 = 0$.

## Data availability

The data supporting this study and its findings are available within the article and Supplementary Information. Data of this study are available from the corresponding author on request. Source data are provided with this paper.

## Code availability

The codes that support the findings of this study are available from the corresponding author upon request.

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

## Acknowledgements

G.H. acknowledges the support of the Air Force Office of Scientific Research under Grant No. AF 9550-18-1-0342 and AF 9550-20-1-0279 with Program Manager Dr. Gregg Abate. A.A. is supported by the NSF Science and Technology Center 'Frontiers of Sound' and the Simons Foundation.

## Author contributions

S.W., N.S. and H.Q. designed the metabeam with integrated circuits and conducted the simulations and measurements. S.W. and H.C. developed the theoretical framework, with contributions from Q.W., A.A. and G.H. S.W., J.C., A.A., H.D. and G.H. interpreted and presented the experimental data. S.W. and G.H. led the writing of the paper and the Supplementary Information, with input from all authors. The problem was conceived by S.W., A.A., and G.H., and the project was supervised by G.H.

## Competing interests

The authors declare no competing interests.
