## [Transparent Peer Review file · Nature Communications]

Experimental Realization of Temporal Refraction and Reflection in Elastic Beams

Corresponding Author: Professor Guoliang Huang

Version 0:

Reviewer comments:

Reviewer #1

(Remarks to the Author)

The authors proposed a temporal refraction and reflection of elastic waves in modulated mechanical metabeams. This manuscript addresses a topic of significant interest in the fields of physics and mechanical engineering and is supported by detailed validation. However, given the broad scope of Nature Communications, there is some question as to whether publishing such a highly specialized article aligns well with the journal's focus. Therefore, along with the concerns outlined in the comments below, the reviewer believes that this manuscript needs a major revision to clarify the novelty of the work and explain the supporting grounds.

1. The reviewer has significant concerns regarding the novelty of the concept and the demonstration methods presented in this paper. The discussion of temporal refraction and reflection in the article appears to offer little distinction from the study, 'What is the temporal analog of reflection and refraction of optical beams? B.W. Plansinis et al., Phys. Rev. Lett. 115, 183901 (2015)'. Moreover, the exploration and implementation of physical phenomena through time-modulation of elastic waves seem to lack unique differentiation compared to the studies published the same research group, such as 'Time-periodic stiffness modulation in elastic metamaterials for selective wave filtering: theory and experiment. G. Trainiti et al., Phys. Rev. Lett. 122, 124301 (2019)', 'Dynamic phononic crystals with spatially and temporally modulated circuit networks. Q. Wu et al., Acta Mech. Sin. 39, 723007 (2023)', and 'Inherent temporal metamaterials with unique time-varying stiffness and damping. Z. Liu et al., Adv. Sci. 11, 2404695 (2024)'. The authors should clearly convey the novelty, superiority, and new physical phenomena introduced in this work, ensuring that readers can readily appreciate the contribution of the study.

2. The reviewer finds it difficult to understand the authors' claim that "energy is not conserved, but momentum is conserved." Based on the reviewer's understanding, the change in resistance leads to a variation in dispersion characteristics (as shown in Figure S1), which, in turn, modulates elastic waves through changes in effective stiffness. However, the assertion that momentum is conserved—implying that the wavevector remains unchanged—seems inappropriate in a scenario where the dispersion relationship itself is altered. Furthermore, in a system where electrical energy is applied as a gain, the claim that energy is not conserved, without a thorough consideration of this aspect, raises questions about the completeness of the physical interpretation. The authors are strongly encouraged to address these concerns and provide a clear explanation to resolve these ambiguities.

3. In Figure 3, the authors present both theoretical and measurement data for various wave phenomena, but it may be difficult to conclude that the two results are in good agreement. The reviewer believes this discrepancy likely arises from the use of the Euler–Bernoulli beam theory to analytically model a beam structure with a relatively small slenderness ratio. To address this issue, the reviewer suggests that applying the Timoshenko beam theory, which accounts for shear deformation and rotational inertia, could provide a more accurate analytical solution and improve the prediction of measurement results.

4. In Figure 2, the spacetime diagram suggests that the measured and simulated results might not have been derived under identical input conditions. The authors state that a 3-cycle tone-burst wave was generated, but it appears that the wave packets in Figures 2a and 2c differ in shape (or possibly due to adjustments in the colormap). Furthermore, in Figure 2f, while the normalized signal of the incident wave is 1, the sum of the refracted and reflected waves does not appear to equal 1. The authors are requested to provide a detailed explanation addressing these discrepancies.

5. Elastic waves propagating in a medium are not limited to flexural waves. However, the authors have focused solely on the

behavior of the slowest flexural wave. Are the effects of other wave modes, such as longitudinal waves, shear waves, or higher-order bending wave modes, not considered in this study? If these considerations fall outside the scope of this study, the authors are encouraged to provide evidence demonstrating that the influence of these additional modes is negligible.

Reviewer #2

(Remarks to the Author)

This manuscript presents research on time-varying metabeams and highlights the significance of design and control over spatial- and time-varying characteristics. The study addresses an important topic and is certainly intriguing. As mentioned by the authors in the introduction, the exploration of temporal interfaces in elastic media is in its infancy, which underscores the necessity for further research in this area—a claim with which I wholeheartedly agree. Overall, the manuscript is technically well-written; however, I believe there are several points that need to be addressed regarding novelty. Therefore, I recommend major revisions to improve these aspects before re-evaluating the manuscript for potential publication in Nature Communications.

- Implementing temporal interfaces in elastic media is challenging. This study experimentally demonstrates this implementation before validating it through numerical simulations. However, it is unclear what specific challenges this research overcomes and what unique strategies have been applied to achieve this. While the implementation of a temporal interface is undoubtedly successful, the experimental results derived from toggling an analog switch raise questions about the experimental design, particularly the basis for obtaining the results displayed in Figures 1b, 1c, and 1d. It is necessary to present a more convincing argument on how these results stem from the research's unique strategy.
- I suggest revising the title. The content following the colon indicates both theory and physical observation. However, the findings of this study begin with experimental results, followed by validation against numerical simulations, and then some confirmation concerning the theoretical aspects. Thus, it may not be appropriate to claim that the paper presents a comprehensive theory. Please consider a title without a colon that accurately reflects the key contributions of the manuscript.
- The titles of the main manuscript and the Supporting Information do not match, and the numbering of authors' affiliations is inconsistent.
- If we define metamaterials as artificial structures that demonstrate unconventional material properties or behaviors, I would appreciate a more convincing emphasis on why the beam in this study qualifies as a "meta" beam. Simply modulating wave propagation with a PZT-attached array may not inherently be considered "meta."
- The experiments presented in the early figures were conducted at frequencies of 6–10 kHz, while the smart waveform morphing in Figure 6 selected 33 kHz. Please address the rationale behind choosing each frequency.
- The manuscript transitions from experimental results in the early sections, which are validated by numerical analyses, to discussing smart waveform morphing and information coding, which are initially designed and then validated through numerical results followed by experimental confirmation. Presenting the early study as yielding results that "happen to provide good results" does not create a natural narrative connecting the experimental outcomes to numerical simulations and theory. This section could benefit from a more meaningful and cohesive presentation.

Reviewer #3

(Remarks to the Author)

The paper addresses temporal refraction and reflection in a metabeam with time varying stiffness. In my view, there are three main contributions to the field: (A) experimental observation of temporal refraction, and corresponding Snell and Fresnel relations, for a continuous elastic medium (B) numerical simulations and theory showing potential applications in the form frequency and amplitude manipulation and (C) experiment and simulation showing an application in information coding (Morse code). The paper is well-written and clear for the most part. The SM is quite large, and one has to refer quite often to it, decreasing readability somewhat. Overall, however, the results are well explained, the analytical calculations seem to be correct and I have no concerns about the validity of the experimental data. There is a lot of interest in systems with time varying properties, and thus the paper could have a large audience in physics, material science, applied math, and more. I have a concern regarding novelty, addressed below, and a number of small suggestions to improve the manuscript. If they are addressed, the article may be suitable for publication.

1) My concern in terms of impact on the field, is that there is significant overlap in contribution (A) of their paper with the paper [48] in the bibliography. Experimental observation of temporal refraction of elastic waves, and corresponding Snell and Fresnel relations are also studied [48]. The platform is different than in the present paper, (continuous media vs discrete media). In my opinion that alone does not justify publication in Nature Communications. On the other hand, this paper also includes item (B) and (C) above, which brings the system closer to potential applications. One could also argue that the platform studied by the authors is more amenable to application (a beam vs an array of magnets). While on page 2 it is written "a temporal interface in elastic continuum medium ... remains explored", I believe a more detailed justification of the work's novelty is needed. Moreover, line 27 claims that the elastic counterpart of Snell's law and Fresnel are unexamined,

yet these were explored in [48].

minor comments:

- 1.) There needs to be a discussion on damping in the maintext, and why it is ignored in the model. Presumably for longer beams, as in the simulations with longer length starting on page 7), damping could play a significant role.
- 2) Should the x axis label for fig. 2h be wavenumber rather than frequency?
- 3) When discussing amplification, the role of nonlinearity will become important, which is not addressed. Can the authors quantify/justify, based on the amplitude of the relevant signals, that ignoring the nonlinearity is justifiable, especially when application is involved?
- 4) Are all simulations, with the exception of stiffness determination, performed on Eq (1)? What algorithm is used for simulation?
- 5) In the SM section 4A, the first sentence states "Fig 2 of main text only experiments are shown", whereas the caption of Fig 2b is apparently simulated. Can the authors be more clear in the caption of Fig 2 of the main text what is simulation and what is experiment, and correct the text in the SM accordingly?
- 6) There is not much detail or explanation to where the transfer function in Fig 7 comes from. In general the three amplitudes needed for the morse coding, why 5.2, 5.4 and 5.3 for the transfer function? How sensitive is the response to the choice of these amplitudes?
- 7) In Fig 7, the amplitudes between the dot and space and dot and dash are somewhat close to each other. What are the associated error bars with the experiment? How can it be justified that the variation between experimental runs is not greater than the difference in the amplitude of the dash,space and dot signals?

Version 1:

Reviewer comments:

Reviewer #1

(Remarks to the Author)

In the revised manuscript, the authors have addressed most of the issues I raised and have significantly improved the clarity of their manuscript. However, the most critical issue still remains unresolved. The electrically shunted elastic beam used for demonstrating temporal refraction and reflection is an experimental platform that the authors have already employed in many of their previous works. While I certainly agree that using the same experimental platform can be a valuable tool for physical interpretation, if the main contribution of this study is to experimentally validate previously proposed physical phenomena (to the extent that the title claims 'Experimental Realization'), then the experimental setup itself must also exhibit a degree of novelty.

Reviewer #3

(Remarks to the Author)

I have read the response and new manuscript. The authors have addressed my concerns. I recommend publication.

Response to the Reviewers' Comments

Reviewer 1

The authors would like to take this opportunity to thank the reviewer for their insights and suggestions. We have responded to the comments below, and the paper has been modified accordingly. All the revised and added parts have been highlighted in red in the response file, revised manuscript, and Supplementary Information.

Comment 1 (overall): The authors proposed a temporal refraction and reflection of elastic waves in modulated mechanical metabeams. This manuscript addresses a topic of significant interest in the fields of physics and mechanical engineering and is supported by detailed validation. However, given the broad scope of Nature Communications, there is some question as to whether publishing such a highly specialized article aligns well with the journal's focus. Therefore, along with the concerns outlined in the comments below, the reviewer believes that this manuscript needs a major revision to clarify the novelty of the work and explain the supporting grounds.

Response: We thank the reviewer for acknowledging the technical value and detailed validation of our study. After carefully reviewing the existing literature, we explicitly highlight both the novelty and broader significance of our work in the following three key aspects:

(I) Physical Realization of Temporal Interfaces in Elastic Beams: Temporal interfaces—abrupt temporal changes in material properties—are extensively explored in optics, but their physical realization in elastic continua remains challenging, primarily due to the stringent requirements of sub-microsecond synchronization. Experimental demonstrations in discrete mass-spring systems (e.g., magnet-based lattices [49]) exhibit time refraction but operate at low working frequencies (~ 10 Hz), resulting in slow switching time (~ 1 ms), and cannot be extended to continuum systems due to the slow response of electromagnetic actuation. **On an experimental level**, (i) we present the first observation of a sharp temporal interface in an elastic beam via piezoelectric patches with circuit-driven modulation at high switching time ($\sim 0.1 \mu\text{s}$), enabling clear observation of temporal refraction and reflection of flexural waves; and (ii) we provide a quantitative analysis of the required transition time for property modulation to achieve ideal refraction and reflection at various operating frequencies (~ 10 kHz).

(II) Field-Theoretic Formulation of Temporal Refraction in Elasticity: Although temporal Snell's law and Fresnel equations are well established in optics, their elastic counterparts lack a field-based formulation. **On a fundamental level**, this study advances the theory of elastodynamics by (i) deriving temporal Snell's law and Fresnel equations for elastic beams directly from the equations of motion, and (ii) formulating momentum conservation and the breakdown of energy conservation in elastic systems with temporal interfaces using Noether's theorem. The resulting framework is general and offers a foundation applicable to phononic, photonic, and quantum systems under temporal modulation.

(III) Engineering Waves via Temporal Interfaces: Unlike in optics, where research has advanced toward a range of wave manipulation strategies, little to no work has successfully leveraged temporal interfaces for elastic wave engineering. **On the application level**, we establish a design principle that enables simultaneous, broadband control of wave attributes—frequency, amplitude, and phase—through temporal interfaces. The resulting wave manipulation capabilities include (i) programmable frequency shifting and amplitude modulation; (ii) anti-reflection functionality via phase matching; and (iii) mechanical information encoding (e.g., Morse code). These features open new avenues for elastic signal processing, energy routing, and programmable mechanical metamaterials.

In light of these three aspects, we have revised the final two paragraphs of the revised main text to highlight the novelty and broader relevance of the study.

“Among these phenomena, wave refraction and reflection at temporal interfaces, are considered one of the most fundamental phenomena, at the foundations of time crystals, yet they are often challenging to achieve in practical wave systems. One of the key challenges is the creation of a temporal boundary, which typically requires a spatially uniform, ultrafast, and large change of wave impedance [19,48]. The associated experimental challenges keep the experimental study of wave scattering at temporal interfaces in its infancy, particularly in the context of elastic media. Recently, the observation of temporal refraction and reflection is reported for electromagnetic waves leveraging transmission-line metamaterials [19,21]. Building on this, temporal interfaces are demonstrated in discrete elastic systems using repelling magnets and electromagnetic control [49], but such platforms cannot be extended to continuum systems due to the slow response of electromagnetic actuation. In contrast, prior studies in elastic continua demonstrate smooth temporal stiffness-damping modulation for inducing amplitude modulation and spectral shaping [50], and periodic modulation for k -space bandgaps [37] and dynamic phononic crystal [51]. However, realizing a sharp and spatially uniform temporal interface remains elusive in continuum elastic media, primarily due to the strict requirements of sub-microsecond synchronization and uniform stiffness modulation across the entire structure. Although temporal Snell’s law and Fresnel equations are well established in optics and discrete elastic systems, no field-based formulation exists for continuum beams. Likewise, momentum conservation and the breakdown of energy conservation—concepts rooted in Noether’s theorem—are not formulated in time-varying beam theory. Furthermore, the use of sharp temporal interfaces for simultaneous and broadband manipulation of multiple wave attributes—frequency, amplitude, and phase—remains largely unexplored in elastic media.

Elastic beams, equipped with piezoelectric patches connected to digital and analog circuits, provide an excellent platform to achieve unconventional elastic wave phenomena, including the non-Hermitian skin effect [52], odd mass density [53], Willis responses [54], frequency conversion [55], and topological pumping [56]. In this study, we report the first experimental realization of a sharp temporal interface in an elastic beam, enabled by real-time, circuit-driven stiffness modulation at the sub-microsecond scale using piezoelectric patches. This setup allows the direct observation of temporal refraction and reflection of flexural waves. We further derive temporal Snell’s law and Fresnel equations for flexural waves directly from the governing equations of motion and formulate momentum conservation and the breakdown of energy conservation in time-varying elastic systems using Noether’s theorem. These theoretical developments are further validated through numerical simulation and experimental observation. By introducing multiple temporal interfaces, we demonstrate further control over the manipulation of flexural waves in both amplitude and frequency spectra. Finally, by programming a smooth time-varying transfer function to realize adiabatic stiffness modulations, we demonstrate additional capabilities in shaping the time-scattered waves in periodic and aperiodic fashion for smart waveform morphing and information coding. These results not only establish a programmable platform for manipulating elastic waves in practical engineering systems, but also deepen the fundamental understanding of wave-matter interactions under temporal modulation.”

Comment 1.1: The reviewer has significant concerns regarding the novelty of the concept and the demonstration methods presented in this paper. The discussion of temporal refraction and reflection in the article appears to offer little distinction from the study, ‘What is the temporal analog of reflection and refraction of optical beams? B.W. Plansinis et al., Phys. Rev. Lett. 115, 183901 (2015)’. Moreover, the exploration and implementation of physical phenomena through time-modulation of elastic waves seem to lack unique differentiation compared to the studies published the same research group, such as ‘Time-periodic stiffness modulation in elastic metamaterials for selective wave filtering: theory and experiment. G. Trainiti et al., Phys. Rev. Lett. 122, 124301 (2019)’, ‘Dynamic phononic crystals with spatially and temporally modulated circuit networks. Q. Wu et al., Acta Mech. Sin. 39, 723007 (2023)’, and ‘Inherent temporal metamaterials with unique time-varying stiffness and damping. Z. Liu et al., Adv. Sci. 11, 2404695 (2024)’. The authors should clearly convey the novelty, superiority, and new physical phenomena introduced in this work, ensuring that readers can readily appreciate the contribution of the study.

Response: We thank the reviewer for this valuable comment and for pointing out the importance of clarifying the novelty and contribution of our work relative to previous studies. To address this concern, we have explicitly distinguished the present work from the cited references. The key differences are summarized below:

- **Compared to Plansinis et al. (PRL 2015):** Plansinis et al. provided a purely theoretical study of temporal refraction in optics, deriving a Snell-like relation based on wavevector conservation. In contrast, our work develops a field-based framework for temporal refraction in continuous elasticity and experimentally demonstrates it for the first time—an achievement that is both technically challenging and practically significant.

Moreover, while Plansinis et al. did not derive Fresnel-like amplitude relations, we derive both Snell and Fresnel analogs from the governing field equations, enabling amplitude-level control. Their treatment of momentum is rooted in the quantum expression $p = \hbar k$, which lacks direct applicability in classic elasticity. By contrast, we derive momentum conservation and breakdown of energy conservation in time-varying elastic continua using Noether’s theorem, yielding physically meaningful quantities for structural wave analysis and mechanical signal processing.

This reference has now been added to the revised manuscript as Ref. [23] and is cited in the Introduction (lines 10–11) as part of the grouped citation “...reflection and refraction^{18–23}...”, to appropriately acknowledge prior foundational work.

- **Compared to Trainiti et al. (PRL 2019):** While both studies employ time-modulated beams using piezoelectric shunting, the physical phenomena explored are fundamentally different. Trainiti et al. investigated frequency-domain effects through **periodic modulation**—such as κ -space bandgaps and narrowband amplification—whereas our study focuses on time-domain temporal scattering to observe the refraction and reflection of elastic waves at sharp temporal interfaces, where the periodic modulation is not necessary. As a result, the modulation switching time in Trainiti et al. (PRL 2019) is not required to be at sub-microsecond synchronization, which is one of significant contribution in our study. In addition, the paper by Trainiti et al. aims for spectral filtering and parametric gain, while our study enables real-time manipulation of waveform frequency, amplitude, and phase, offering a programmable foundation for elastic signal processing and information encoding.

To clarify the distinction between our work and that of Trainiti et al. (Ref. [37]), we have revised Lines 24–26 of the revised manuscript accordingly

“In contrast, prior studies in elastic continua demonstrate smooth temporal stiffness-damping modulation for inducing amplitude modulation and spectral shaping [50], and periodic modulation for k -space bandgaps [37] and dynamic phononic crystal [51].”

- **Compared to Wu et al. (Acta Mech. Sin. 2023):** This prior study from our group also uses piezoelectric shunting in beams through the **smooth sinusoidal modulation** to induce frequency bandgaps and spectral filtering in the frequency domain. In contrast, our current study focuses on sharp temporal interfaces at sub-microsecond synchronization that induce time-domain wave splitting, enabling the observation of temporal refraction and reflection—a qualitatively different class of dynamics requiring distinct theoretical tools. The smooth modulation by Wu et al. (Acta Mech. Sin.2023) will not induce the temporal refraction and reflection. Moreover, beyond hardware differences (e.g., sub-microsecond synchronization), our study also expands functionality—demonstrating waveform morphing, anti-reflection, and information encoding.

To clarify the distinction between our work and that of Wu et al. (Ref. [51]), we have revised Lines 24–26 of the revised manuscript accordingly

“In contrast, prior studies in elastic continua demonstrate smooth temporal stiffness-damping modulation for inducing amplitude modulation and spectral shaping [50], and periodic modulation for k -space bandgaps [37] and dynamic phononic crystal [51].”

- **Compared to Liu et al. (Adv. Sci. 2024):** While both works involve programmable modulation in piezoelectric beams, the focus and physical mechanisms are fundamentally different. The study by Liu et al. (Adv. Sci. 2024), collaborated with our group, targets amplitude modulation and spectral shaping via **smooth stiffness-damping modulation** in the frequency domain, which limits to adiabatic changes only. In contrast, our present work explores temporal refraction and reflection, which need abrupt stiffness modulation at sub-microsecond synchronization in the time domain.

Moreover, the previous platform is governed by coupled electrical ODEs and a mechanical PDE, making temporal scattering analysis intractable. Our system, by contrast, realizes a pure mechanical beam with time-varying stiffness, governed by a PDE, enabling clear interpretation of temporal interfaces. In terms of functionality, their approach supports amplitude modulation only, achieved through damping. In contrast, we demonstrate control over amplitude, frequency, and phase.

To clarify the distinction between our work and that of Liu et al. (Ref. [50]), we have revised Lines 24–26 of the revised manuscript accordingly

“In contrast, prior studies in elastic continua demonstrate smooth temporal stiffness-damping modulation for inducing amplitude modulation and spectral shaping [50], and periodic modulation for k -space bandgaps [37] and dynamic phononic crystal [51].”

We note that the novelty and broader relevance of this work are addressed in detail in our response to Comment 1 (Overall). Here, we provide additional clarification regarding its **novelty, superiority, and the new physical phenomena** demonstrated.

- **Platform novelty — continuous elastic medium with sharp temporal interfaces in the time-domain:** Realizing temporal interfaces in continuum elastic systems poses significant challenges due to the high operating frequency (~ 10 kHz) and wave speed (~ 1 km/s), which demand sub-microsecond, spatially uniform modulation to create a sharp temporal boundary. Our analysis shows that the stiffness must change within less than $1/100$ of the wave period to achieve effective refraction and reflection. We implement this by using analog switches with ~ 0.1 μ s response time, synchronized beam via a centralized 180 MHz microcontroller.
- **Theoretical superiority — field-theoretic momentum formulation from Noether’s theorem:** We develop a momentum conservation law based on *field-theoretic principles* by applying Noether’s theorem to the Euler–Bernoulli beam with time-dependent stiffness. This derivation goes beyond the typical ray or band-structure arguments used in optics and acoustics, and enables a rigorous analysis of momentum transport in time-varying elastic continua. In particular, we distinguish between *canonical momentum* (e.g., $\rho A w_t$) and the *Noether momentum*, which arises from spatial translation invariance and takes the form of a conserved flux: $\int \rho A w_t w_x dx$.

In systems with temporal modulation, such as when $D(t) \propto 1/t^2$, the Lagrangian has explicit time dependence, and the solution is $w \propto \sqrt{t}$. As a result, the canonical momentum $\rho A w_t$ is *not* conserved—even for a plane wave—because energy input or extraction varies over time. However, the Noether momentum, derived from spatial symmetry, remains conserved. Here, $w_t \propto 1/\sqrt{t}$ and $w_x \propto \sqrt{t}$, their product yields a time-independent momentum flux density, ensuring conservation of the total Noether momentum even though the canonical momentum varies. This subtle distinction is critical for time-varying systems, where flux conservation, not pointwise invariance, determines physical response.

By making this distinction explicit, our formulation provides a practical tool for analyzing *energy transfer, momentum exchange, and wave scattering* in systems with temporal interfaces. It also enables consistent definitions of reflection and refraction in the time domain (via Fresnel-like laws), and establishes a theoretical basis that can be extended to more complex spatiotemporally modulated systems, including phononic, photonic, and quantum platforms.

- **New physical phenomena — time-domain Snell/Fresnel analogs and waveform manipulation in elastic beams:** This work demonstrates a class of *real-time wave phenomena* in continuous elastic media,

including temporal refraction, reflection, and energy redistribution governed by field-derived analogs of Snell’s and Fresnel’s laws. In contrast to prior studies—such as Liu et al., which focus on slow amplitude modulation and spectral shaping via stiffness–damping modulation, and Trainiti et al., which explore k -space bandgaps via periodic modulation—our system directly observes time-domain wavefront splitting and transient waveform reshaping at sharp temporal interfaces.

These effects go beyond steady-state filtering or gain, enabling dynamic control over amplitude, frequency, and phase. In particular, we demonstrate programmable waveform morphing, anti-reflection through phase tuning, and mechanical information encoding (e.g., Morse code), establishing temporal interfaces as a practical mechanism for elastic signal processing and programmable mechanical systems.

We believe these clarifications sufficiently address the reviewer’s concern regarding novelty, superiority, and new physical phenomena. Corresponding revisions are made in the main text and detailed in our response to Comment 1.

Comment 1.2: The reviewer finds it difficult to understand the authors’ claim that “energy is not conserved, but momentum is conserved.” Based on the reviewer’s understanding, the change in resistance leads to a variation in dispersion characteristics (as shown in Figure S1), which, in turn, modulates elastic waves through changes in effective stiffness. However, the assertion that momentum is conserved—implying that the wavevector remains unchanged—seems inappropriate in a scenario where the dispersion relationship itself is altered. Furthermore, in a system where electrical energy is applied as a gain, the claim that energy is not conserved, without a thorough consideration of this aspect, raises questions about the completeness of the physical interpretation. The authors are strongly encouraged to address these concerns and provide a clear explanation to resolve these ambiguities.

Response: We thank the reviewer for this insightful comment. The statement that “energy is not conserved, but momentum is conserved” is a well-known conclusion of time-metamaterials [18-23] and it stems directly from preserved translation invariance in space. Since translational invariance is preserved (we change the elastic properties uniformly in space), we must preserve momentum and wave number. This finding, grounded in the fundamental relationship between symmetry and conservation laws as formalized by Noether’s theorem, is well-established in the optics literature, but it is new in its elastic counterpart. As shown in Supplementary Section 13, we derive the energy relations in Eq. (S28)

$$H = \int \mathcal{H} dx = \int dx \left(\frac{1}{2} \rho A \partial_t w^\dagger(x, t) \partial_t w(x, t) + \frac{1}{2} EI \partial_{xx} w^\dagger(x, t) \partial_{xx} w(x, t) \right) \quad (1)$$

and momentum in Eq. (S41)

$$P = \int p dx = \int dx \rho A [(\partial_t w^\dagger) w_x + (\partial_t w) w_x^\dagger] \quad (2)$$

based on the time-dependent Lagrangian given in Eq. (S14)

$$\mathcal{L} = \frac{1}{2} \rho A \partial_t w^\dagger(x, t) \partial_t w(x, t) - \frac{1}{2} EI \partial_{xx} w^\dagger(x, t) \partial_{xx} w(x, t) \quad (3)$$

In our system, the Lagrangian is explicitly time-dependent (due to stiffness modulation), which breaks time-translation symmetry and leads to the non-conservation of energy in Eq. (S31)

$$H_0 \neq H_1 \quad (4)$$

where the energy before temporal interface H_0 and the energy after temporal interface H_1 are

$$H_0 = \frac{1}{Z_0^2} \rho A A_i^2, \quad H_1 = \frac{Z_0^4 + Z_0^2 Z_1^2}{2Z_1^4} H_0, \quad (5)$$

respectively. However, it preserves spatial translation symmetry, which ensures conservation of momentum in the

FIG. S13. Refraction and reflection of flexural waves at a time interface. **a** (c) Spacetime diagram of wave refraction and reflection at a spatial (temporal) interface. The subscripts i , r , and t correspond to the incident, reflected, and refracted waves, respectively. **b** (d) The top panel shows the index of refraction versus x (t), with an abrupt change at x_1 (t_1). The bottom panel displays the dispersion curves: the medium before (after) x_1 (t_1) is represented by a purple (orange) solid line. The blue, red, and yellow points correspond to the incident, reflected, and refracted waves, respectively.

Noether sense in Eq. (S44)

$$P_0 = P_1 = 2Z_0 k_0^3 A_i^2. \quad (6)$$

To further clarify this, we refer the reviewer to Fig. S13(c,d) [2,17]. In the bottom panel of Fig. S13(d), we illustrate the dispersion relation before (purple) and after (orange) the temporal interface. The incident wave (purple arrow in Fig. S13(c)) lies on the initial dispersion curve (the purple dot in Fig. S13(d)) and, after crossing the temporal interface, splits into a reflected wave (blue arrow) and a refracted wave (orange arrow). Their corresponding (k, ω) values lie on the new dispersion branch (bottom panel of Fig. S13(d)), but all share the same wavenumber k , confirming momentum conservation which is proportional to k in Eq. (2). However, the frequency ω differs, leading to a shift in energy (which scales with ω^2 in Eq. (1)). Thus, energy is not conserved, while momentum is.

Regarding the role of the electrical circuit: we agree with the reviewer that, if the full coupled electromechanical system is considered, energy is conserved globally. However, in our analysis, we intentionally focus on the *mechanical beam* as a subsystem of interest. The electrical components act as an engineered environment that dynamically modulates the mechanical properties (i.e., the stiffness $D(t)$) of the beam. From this perspective, energy is exchanged between the mechanical and electrical domains, and hence mechanical energy alone is not conserved. The energy is transferred from the modulation network to the wave.

This perspective reflects a widely adopted modeling paradigm in the study of open systems in physics and engineering, where a subsystem with unconventional properties is engineered by carefully designing its surrounding environment. In our case, the environment—the shunting circuit—is constructed to emulate a time-dependent Euler–Bernoulli beam, a behavior that cannot be realized in passive mechanical systems alone. This leads to an effective model in which the beam behaves as a non-autonomous system with explicitly time-varying stiffness. Within this subsystem, energy is not conserved due to the external modulation, while momentum remains conserved due to preserved spatial translation symmetry—fully consistent with the underlying Noetherian framework.

We have revised Lines 130–145 in the revised main text as follows:

“At a temporal interface, time-translation symmetry is broken, leading to the breakdown of energy conservation according to Noether’s theorem. In Supplementary Section 11, the total energy of the elastic beam is given by

$$H = \int dx \left(\frac{1}{2} \rho A, \partial_t w^\dagger(x, t) \partial_t w(x, t) + \frac{1}{2} EI, \partial_{xx} w^\dagger(x, t) \partial_{xx} w(x, t) \right),$$

where † denotes the Hermitian conjugate. Before the temporal interface, the energy is

$$H_0 = \frac{1}{Z_0^2} \rho A A_i^2,$$

and after the interface it becomes

$$H_1 = \frac{Z_0^4 + Z_0^2 Z_1^2}{2Z_1^4} H_0,$$

which differs from H_0 when $Z_1 \neq Z_0$. In contrast, spatial translation symmetry is preserved across the interface, so momentum remains conserved. This conserved momentum, also known as Noether’s charge, of the elastic beam is given by

$$P = \int \rho A \left[(\partial_t w)^\dagger \partial_x w + \partial_t w (\partial_x w)^\dagger \right] dx.$$

The detailed derivation of Eq. (13) using the complex scalar field theory of the Euler-Bernoulli beam is provided in Supplementary Section 11. The momentum of the wave before the time switching is

$$P_0 = 2\rho A \omega_0 k_0 A_i^2, \tag{7}$$

whereas the momentum of the waves after the switching time is

$$P_1 = 2\rho A \omega_1 k_0 (T^2 - R^2) A_i^2. \tag{8}$$

With the aid of Eq. (9), the conservation of momentum can be easily verified as

$$P_0 = P_1 = 2Z_0 k_0^3 A_i^2. \tag{9}$$

In addition, the momentum of both the incident and scattered waves is proportional to k_0^3 , indicating that the wavenumber remains invariant. It is worth noting that although the full electromechanical system conserves energy, our analysis treats the mechanical beam as an open subsystem. The shunting circuit serves as an engineered environment that modulates stiffness, resulting in mechanical energy exchange and thus energy non-conservation, while momentum remains conserved through spatial translation symmetry.”

Comment 1.3: In Figure 3, the authors present both theoretical and measurement data for various wave phenomena, but it may be difficult to conclude that the two results are in good agreement. The reviewer believes this discrepancy likely arises from the use of the Euler–Bernoulli beam theory to analytically model a beam structure with a relatively small slenderness ratio. To address this issue, the reviewer suggests that applying the Timoshenko beam theory, which accounts for shear deformation and rotational inertia, could provide a more accurate analytical solution and improve the prediction of measurement results.

Response: For a Timoshenko beam, the coupled equations for the transverse displacement $w(x, t)$ and the cross-

sectional rotation $\phi(x, t)$ are given by:

$$\begin{aligned}\rho A \frac{\partial^2 w}{\partial t^2} &= \frac{\partial}{\partial x} \left[\kappa AG \left(\frac{\partial w}{\partial x} - \phi \right) \right], \\ \rho I \frac{\partial^2 \phi}{\partial t^2} &= \frac{\partial}{\partial x} \left(EI \frac{\partial \phi}{\partial x} \right) + \kappa AG \left(\frac{\partial w}{\partial x} - \phi \right),\end{aligned}\tag{10}$$

where ρ is the material density, A is the cross-sectional area, I is the area moment of inertia, E is Young's modulus, G is the shear modulus, and $\kappa = 5/6$ is the shear correction factor.

Assuming harmonic wave solutions of the form:

$$w(x, t) = W e^{i(kx - \omega t)}, \quad \phi(x, t) = \Phi e^{i(kx - \omega t)},\tag{11}$$

the equations reduce to a compact matrix form:

$$\begin{pmatrix} \kappa AG k^2 - \omega^2 \rho A & i \kappa AG k \\ -i \kappa AG k & EI k^2 + \kappa AG - \omega^2 \rho I \end{pmatrix} \begin{pmatrix} W \\ \Phi \end{pmatrix} = \mathbf{0}.\tag{12}$$

Setting the determinant of the coefficient matrix to zero yields the dispersion relation:

$$\frac{\rho^2 AI}{\kappa AG} \omega^4 - \left[\rho A + \left(\rho I + \frac{EI \rho A}{\kappa AG} \right) k^2 \right] \omega^2 + EI k^4 = 0.\tag{13}$$

Solving for ω as a function of k , we obtain the positive real branch:

$$\omega = \sqrt{\frac{\rho A + \left(\rho I + \frac{EI \rho A}{\kappa AG} \right) k^2 + \sqrt{\left[\rho A + \left(\rho I + \frac{EI \rho A}{\kappa AG} \right) k^2 \right]^2 - 4 \frac{\rho^2 AI}{\kappa AG} EI k^4}}{2 \frac{\rho^2 AI}{\kappa AG}}}.\tag{14}$$

The corresponding normalized eigenvector is given by:

$$\psi(\omega, k) = \frac{1}{\sqrt{(\kappa AG k^2 - \omega^2)^2 - (\kappa AG k)^2}} \begin{pmatrix} -i \kappa AG k \\ \kappa AG k^2 - \omega^2 \end{pmatrix},\tag{15}$$

and the complete time-harmonic solution becomes:

$$\Psi(x, t) = \psi(\omega, k) e^{i(kx - \omega t)}.\tag{16}$$

Suppose that at a specific time $t = t_0$, the beam's effective stiffness changes abruptly due to the switching of an external electronic circuit. In our case, this change primarily affects the bending stiffness. We define the effective bending stiffness before and after the temporal interface as:

$$E_0 I \quad \text{and} \quad E_1 I,\tag{17}$$

respectively. The piezoelectric patches do not affect the shear term or the mass parameters. Therefore, we assume that the shear term κAG and the mass parameters ρA and ρI remain unchanged.

Under this condition, Snell's law for the temporal interface can be expressed as:

$$\omega_0 n_0 = \omega_1 n_1,\tag{18}$$

where n_0 and n_1 are the effective refractive indices corresponding to stiffnesses $E_0 I$ and $E_1 I$, respectively, and are computed via Eq. (14).

We assume the wavefield before the temporal interface takes the form:

$$\Psi_0(x, t) = \psi_0(\omega_0, k) e^{ikx - i\omega_0 t}, \quad (19)$$

where $\psi_0(\omega_0, k)$ is the eigenvector in Eq. (15) for frequency ω_0 . While the wavefield after the interface contains both forward- and backward-propagating components:

$$\Psi_1(x, t) = T \psi_1(\omega_1, k) e^{ikx - i\omega_1 t} + R \psi_1(\omega_1, k) e^{ikx + i\omega_1 t}. \quad (20)$$

where $\psi_1(\omega_1, k)$ is the eigenvector in Eq. (15) for frequency ω_1 , T is the refracted coefficient, and R is the reflected coefficient. At the temporal interface $t = t_0$, the continuity conditions for displacement and velocity are:

$$\Psi_0 = \Psi_1, \quad \partial_t \Psi_0 = \partial_t \Psi_1. \quad (21)$$

Substituting the waveforms into these conditions and evaluating at $t = t_0$, we obtain:

$$\begin{aligned} \psi_0(\omega_0, k) &= T \psi_1(\omega_1, k) + R \psi_1(\omega_1, k), \\ \omega_0 \psi_0(\omega_0, k) &= \omega_1 T \psi_1(\omega_1, k) - \omega_1 R \psi_1(\omega_1, k). \end{aligned} \quad (22)$$

We now take the inner product of both equations with the dual (left) eigenvector $\psi_1^\dagger(\omega_1, k)$. This yields:

$$\begin{aligned} \psi_1^\dagger(\omega_1, k) \psi_0(\omega_0, k) &= T + R, \\ \omega_0 \psi_1^\dagger(\omega_1, k) \psi_0(\omega_0, k) &= \omega_1 (T - R). \end{aligned} \quad (23)$$

Solving this system of equations, we obtain the Fresnel-type refraction and reflection coefficients:

$$\begin{aligned} T &= \frac{\omega_1 + \omega_0}{2\omega_0} \psi_1^\dagger(\omega_1, k) \psi_0(\omega_0, k), \\ R &= \frac{\omega_1 - \omega_0}{2\omega_0} \psi_1^\dagger(\omega_1, k) \psi_0(\omega_0, k). \end{aligned} \quad (24)$$

These expressions generalize the Fresnel equations to the Timoshenko beam case, where the reflection and refraction amplitudes are determined by both frequency shift and mode overlap across the temporal interface.

We now turn to the numerical evaluation of Snell's law and Fresnel coefficients using the Timoshenko beam model, and compare the results with those obtained from the Euler–Bernoulli beam model and COMSOL simulation. As shown in Fig. 2, the dispersion curves from all three models closely match, confirming the validity of the Euler–Bernoulli approximation. When the bending stiffness changes from $D/D_0 = 1$ to $D/D_0 = 0.85^2$, the normalized frequencies for the Timoshenko beam, Euler–Bernoulli beam, and COMSOL simulation change from 0.991, 0.989, 1 to 0.844, 0.842, 0.85, respectively. This yields corresponding ratios ω_1/ω_0 of 0.8517, 0.8514, and 0.850, respectively.

Moreover, the mode overlap term $\psi_1^\dagger(\omega, k) \psi_0(\omega, k)$, which appears in the Fresnel equations (Eq. 24), evaluates to 0.999999311089990, indicating that the mode shapes before and after switching remain nearly identical. Therefore, the reflection and refraction coefficients predicted by the Timoshenko and Euler–Bernoulli models are virtually indistinguishable. The differences are negligible in practical terms.

This agreement can be understood by introducing the characteristic shear length scale

$$\ell = \sqrt{\frac{EI}{\kappa GA}}, \quad (25)$$

which leads to the dimensionless dynamic slenderness parameter $\Lambda = \ell k$. In our setup, $\Lambda \sim 10^{-4}$, indicating that shear effects are minimal and the Euler–Bernoulli model is valid.

Fig. R1.4. Dispersion curves obtained from the Timoshenko beam model, the Euler–Bernoulli model, and COMSOL simulation. The top branch corresponds to the case with the circuit switch turned off. The bottom branches correspond to circuits with $R_1 = 5 \text{ k}\Omega$ ($D/D_0 = 0.85^2$, used in the experiment) and $R_1 = 5.6 \text{ k}\Omega$ ($D/D_0 = 0.25$, used in Fig. S7(a)).

To further validate this conclusion, we consider a more extreme case where the stiffness drops from $D/D_0 = 1$ to $D/D_0 = 0.25$ at the temporal interface. The normalized frequencies change from 0.991, 0.989, 1 to 0.498, 0.497, 0.5 for the Timoshenko beam, Euler–Bernoulli beam, and COMSOL, respectively, yielding $\omega_1/\omega_0 = 0.5025$, 0.5025, and 0.500. The mode overlap in this case is still very high: $\psi_1^\dagger(\omega, k)\psi_0(\omega, k) = 0.999998443354908$. Again, the Fresnel coefficients predicted by both beam models are nearly identical.

These results confirm that even under strong modulation, the Euler–Bernoulli beam theory provides sufficiently accurate predictions, and the effects of shear deformation and rotary inertia are negligible for our study. The small discrepancies observed are likely attributed to two factors: (i) incomplete termination of low-frequency incident signals before reaching the temporal boundary, and (ii) possible breakdown of the homogeneous beam assumption at high frequencies due to microstructural effects. These factors, rather than the beam model itself, account for the minor deviations between theory and experiment.

In response to the reviewer’s suggestion, we have added a detailed derivation and analysis of the Timoshenko beam model in the Supplementary Section 11. This addition demonstrates that while the Timoshenko model captures shear and rotary inertia effects, their influence is minimal in our experimental regime. Thus, the use of the Euler–Bernoulli beam theory in the main text is justified for the level of precision required.

To reflect this point in the main text, we have also added a clarifying sentence in Lines 126–128, explicitly noting the use of the Timoshenko beam model in our theoretical analysis:

“A detailed justification for neglecting shear deformation (via the Timoshenko beam model), as well as the negligible influence of higher-order flexural modes, longitudinal modes, damping, and nonlinear effects, is provided in Supplementary Section 11.”

Comment 1.4: In Figure 2, the spacetime diagram suggests that the measured and simulated results might not have been derived under identical input conditions. The authors state that a 3-cycle tone-burst wave was generated, but it appears that the wave packets in Figures 2a and 2c differ in shape (or possibly due to adjustments in the colormap). Furthermore, in Figure 2f, while the normalized signal of the incident wave is 1, the sum of the refracted and reflected waves does not appear to equal 1. The authors are requested to provide a detailed explanation addressing these discrepancies.

Response: Thank you for the careful observation. In Figs. 2a and 2c, we plot the magnitude of the flexural wave field, $|w(x, t)|$, rather than the signed displacement $w(x, t)$. As a result, the number of peaks at position $x = 0$ appears doubled. For example, the number of peaks in the simulation (Fig. 2c) is 6, while in the experiment

(Fig. 2a) it slightly deviates six. Additionally, due to the dispersive nature of flexural waves, the number of visible peaks tends to increase as the wave propagates.

We appreciate the reviewer’s sharp observation regarding the waveform discrepancy in Fig. 2a. During our experiments, we also observed that the number of cycles in the measured waveform slightly deviates from three, despite applying a nominal 3-cycle tone burst. This deviation arises because a bandpass filter (1–40 kHz) is applied within the laser vibrometer to remove low-frequency and high-frequency noise. The bandpass filtering process slightly alters the waveform shape in the time domain but does not affect the amplitude distribution in the frequency domain within our frequency range of interest (6–10 kHz). Therefore, this minor inconsistency in the time-domain waveform does not impact the main conclusions regarding Snell’s law, the Fresnel equations, and momentum conservation.

The statement that the sum of the normalized amplitudes of the refracted and reflected waves equals one is valid only in the wavenumber domain for dispersive waves²⁴. In Fig. 2f, the sum of the normalized amplitudes in the frequency domain is not equal to one; however, in the wavenumber domain shown in Fig. 2h, the sum does equal one.

In addition, to clarify the plotted quantity, we have added a statement in the main text (Line 66-67) indicating that the magnitude of the flexural wave field, $|w(x, t)|$, is measured and plotted in Fig. 2a:

“The magnitude of flexural wave field, $|w(x, t)|$, is measured throughout the system using a scanning laser Doppler vibrometer (Polytec PSV-400), as shown in Fig. 2a.”

Comment 1.5: Elastic waves propagating in a medium are not limited to flexural waves. However, the authors have focused solely on the behavior of the slowest flexural wave. Are the effects of other wave modes, such as longitudinal waves, shear waves, or higher-order bending wave modes, not considered in this study? If these considerations fall outside the scope of this study, the authors are encouraged to provide evidence demonstrating that the influence of these additional modes is negligible.

Response: In our study, the excitation frequency for temporal refraction and reflection is 6–10 kHz, which is significantly lower than the frequencies of the shear mode (776 kHz) and higher-order bending modes (above 80 kHz). As shown in Fig. R1.1(a, b), the dispersion curves confirm that these modes lie well beyond the excitation frequency, and Fig. R1.1(c, d) illustrates their associated mode shapes. Additionally, the incident tone-burst signal amplitude decays to below 0.01 at frequencies above 30 kHz, making the excitation of these higher modes negligible in the experimental testing. In experiments, a 1–40 kHz bandpass filter is also applied to further suppress potential high-frequency or non-flexural contributions.

Regarding longitudinal waves, they can in principle be excited by the piezoelectric actuator. However, the group velocity of the longitudinal mode is nearly eight times that of the fundamental flexural mode, meaning that any longitudinal wave passes through the time-varying segment well before the temporal modulation is active. Therefore, longitudinal waves do not affect the observed wavefields in either experiment or simulation.

To further support this conclusion, we provide time-domain simulations and their Fourier analysis in Fig. R1.1(e, f). The simulated spacetime diagram (identical to Fig. S7(a)) shows a clean refraction/reflection pattern at the temporal interface, without local distortions or fast-propagating components. The frequency–wavenumber spectrum obtained by 2D Fourier transform shows a single dominant branch corresponding to the fundamental flexural mode, with no visible signatures of additional modes. These results provide both physical reasoning and numerical evidence that the influence of higher-order, shear, and longitudinal modes is negligible in the present study.

To address this point, we have included the corresponding analysis at the end of Supplementary Section 11:

“We now turn to discuss the negligibility of other modes. In our study, the excitation frequency for temporal refraction and reflection is 6–10 kHz, which is significantly lower than the characteristic frequencies of shear modes (776 kHz) and higher-order bending modes (above 80 kHz). The dispersion relations and mode shapes confirm that these modes lie far outside the excitation range. Moreover, the

Fig. R1.1. **a, b.** The dispersion curves for the open circuit (**a**) and for the circuit with $R_1 = 5.43 \text{ k}\Omega$ (**b**). **c, d.** The corresponding mode shapes of the first four frequencies of the dispersion curves for the open circuit (**c**) and for the circuit with $R_1 = 5.43 \text{ k}\Omega$ (**d**) at wavenumber $k = 119 \text{ rad/m}$, respectively. **e.** The simulated spacetime diagram of temporal refraction and reflection from the open circuit to $R_1 = 5.43 \text{ k}\Omega$. This is the same as Fig. S7(**a**). **f.** The dispersion curves of the medium before (orange) and after (blue) the switching event, overlaid with a background contour diagram obtained from a 2D Fourier transform of the spacetime data in **e**. This diagram is the same as Fig. S7(**b**) but shown over a larger frequency range.

amplitude of the incident tone-burst signal drops below 0.01 at frequencies above 30 kHz, effectively suppressing the excitation of higher-order modes in practice. In experiments, a 1–40 kHz bandpass filter is also applied to further suppress potential high-frequency or non-flexural contributions.

Regarding longitudinal waves, they can in principle be excited by the piezoelectric actuator. However, the group velocity of the longitudinal mode is nearly eight times that of the fundamental flexural mode, meaning that any longitudinal wave passes through the time-varying segment well before the temporal modulation is active. Therefore, longitudinal waves do not affect the observed wavefields in either experiment or simulation.”

To reflect this point in the main text, we have also added a clarifying sentence in Lines 126–128, explicitly noting that the influence of higher-order, shear, and longitudinal modes is negligible under the current experimental conditions.

“A detailed justification for neglecting shear deformation (via the Timoshenko beam model), as well as the negligible influence of higher-order flexural modes, longitudinal modes, damping, and nonlinear effects, is provided in Supplementary Section 11.”

Reviewer 2

The authors would like to take this opportunity to thank the reviewer for their insights and suggestions. We have responded to the comments below, and the paper has been modified accordingly. All the revised and added parts have been highlighted in red in the response file, revised manuscript, and Supplementary Information.

Comment 2 (Overall): This manuscript presents research on time-varying metabeams and highlights the significance of design and control over spatial- and time-varying characteristics. The study addresses an important topic and is certainly intriguing. As mentioned by the authors in the introduction, the exploration of temporal interfaces in elastic media is in its infancy, which underscores the necessity for further research in this area—a claim with which I wholeheartedly agree. Overall, the manuscript is technically well-written; however, I believe there are several points that need to be addressed regarding novelty. Therefore, I recommend major revisions to improve these aspects before re-evaluating the manuscript for potential publication in Nature Communications.

Response: We thank the reviewer for acknowledging the technical merits of our study and for raising the important question of novelty. We summarize below the main contributions in terms of experimental realization, theoretical formulation, and application potential, with clear distinctions from prior work.

(I) Physical Realization of Temporal Interfaces in Elastic Beams: Temporal interfaces—abrupt temporal changes in material properties—are extensively explored in optics, but their physical realization in elastic continua remains challenging, primarily due to the stringent requirements of sub-microsecond synchronization. Experimental demonstrations in discrete mass-spring systems (e.g., magnet-based lattices [49]) exhibit time refraction but operate at low working frequencies (~ 10 Hz), resulting in slow switching time (~ 1 ms), and cannot be extended to continuum systems due to the slow response of electromagnetic actuation. **On an experimental level**, (i) we present the first observation of a sharp temporal interface in an elastic beam via piezoelectric patches with circuit-driven modulation at high switching time ($\sim 0.1 \mu\text{s}$), enabling clear observation of temporal refraction and reflection of flexural waves; and (ii) we provide a quantitative analysis of the required transition time for property modulation to achieve ideal refraction and reflection at various operating frequencies (~ 10 kHz).

(II) Field-Theoretic Formulation of Temporal Refraction in Elasticity: Although temporal Snell’s law and Fresnel equations are well established in optics, their elastic counterparts lack a field-based formulation. **On a fundamental level**, this study advances the theory of elastodynamics by (i) deriving temporal Snell’s law and Fresnel equations for elastic beams directly from the equations of motion, and (ii) formulating momentum conservation and the breakdown of energy conservation in elastic systems with temporal interfaces using Noether’s theorem. The resulting framework is general and offers a foundation applicable to phononic, photonic, and quantum systems under temporal modulation.

(III) Engineering Waves via Temporal Interfaces: Unlike in optics, where research has advanced toward a range of wave manipulation strategies, little to no work has successfully leveraged temporal interfaces for elastic wave engineering. **On the application level**, we establish a design principle that enables simultaneous, broadband control of wave attributes—frequency, amplitude, and phase—through temporal interfaces. The resulting wave manipulation capabilities include (i) programmable frequency shifting and amplitude modulation; (ii) anti-reflection functionality via phase matching; and (iii) mechanical information encoding (e.g., Morse code). These features open new avenues for elastic signal processing, energy routing, and programmable mechanical metamaterials.

In light of these three aspects, we have revised the final two paragraphs of the revised main text to highlight the novelty and broader relevance of the study.

“Among these phenomena, wave refraction and reflection at temporal interfaces, are considered one of the most fundamental phenomena, at the foundations of time crystals, yet they are often challenging to achieve in practical wave systems. One of the key challenges is the creation of a temporal boundary,

which typically requires a spatially uniform, ultrafast, and large change of wave impedance [19,48]. The associated experimental challenges keep the experimental study of wave scattering at temporal interfaces in its infancy, particularly in the context of elastic media. Recently, the observation of temporal refraction and reflection is reported for electromagnetic waves leveraging transmission-line metamaterials [19,21]. Building on this, temporal interfaces are demonstrated in discrete elastic systems using repelling magnets and electromagnetic control [49], but such platforms cannot be extended to continuum systems due to the slow response of electromagnetic actuation. In contrast, prior studies in elastic continua demonstrate smooth temporal stiffness-damping modulation for inducing amplitude modulation and spectral shaping [50], and periodic modulation for k -space bandgaps [37] and dynamic phononic crystal [51]. However, realizing a sharp and spatially uniform temporal interface remains elusive in continuum elastic media, primarily due to the strict requirements of sub-microsecond synchronization and uniform stiffness modulation across the entire structure. Although temporal Snell’s law and Fresnel equations are well established in optics and discrete elastic systems, no field-based formulation exists for continuum beams. Likewise, momentum conservation and the breakdown of energy conservation—concepts rooted in Noether’s theorem—are not formulated in time-varying beam theory. Furthermore, the use of sharp temporal interfaces for simultaneous and broadband manipulation of multiple wave attributes—frequency, amplitude, and phase—remains largely unexplored in elastic media.

Elastic beams, equipped with piezoelectric patches connected to digital and analog circuits, provide an excellent platform to achieve unconventional elastic wave phenomena, including the non-Hermitian skin effect [52], odd mass density [53], Willis responses [54], frequency conversion [55], and topological pumping [56]. In this study, we report the first experimental realization of a sharp temporal interface in an elastic beam, enabled by real-time, circuit-driven stiffness modulation at the sub-microsecond scale using piezoelectric patches. This setup allows the direct observation of temporal refraction and reflection of flexural waves. We further derive temporal Snell’s law and Fresnel equations for flexural waves directly from the governing equations of motion and formulate momentum conservation and the breakdown of energy conservation in time-varying elastic systems using Noether’s theorem. These theoretical developments are further validated through numerical simulation and experimental observation. By introducing multiple temporal interfaces, we demonstrate further control over the manipulation of flexural waves in both amplitude and frequency spectra. Finally, by programming a smooth time-varying transfer function to realize adiabatic stiffness modulations, we demonstrate additional capabilities in shaping the time-scattered waves in periodic and aperiodic fashion for smart waveform morphing and information coding. These results not only establish a programmable platform for manipulating elastic waves in practical engineering systems, but also deepen the fundamental understanding of wave–matter interactions under temporal modulation.”

Comment 2.1: Implementing temporal interfaces in elastic media is challenging. This study experimentally demonstrates this implementation before validating it through numerical simulations. However, it is unclear what specific challenges this research overcomes and what unique strategies have been applied to achieve this. While the implementation of a temporal interface is undoubtedly successful, the experimental results derived from toggling an analog switch raise questions about the experimental design, particularly the basis for obtaining the results displayed in Figures 1b, 1c, and 1d. It is necessary to present a more convincing argument on how these results stem from the research’s unique strategy.

Response: We thank the reviewer for the insightful comments. Figures 1b–d illustrate the design of the time-varying stiffness profiles used in our system, rather than direct scattering results. These curves are obtained from COMSOL simulations, as detailed in Supplementary Section 1, and correspond to the input $R_1(t)$ signals used to implement the temporal modulation. They capture the key features enabled by our experimental strategy, including sharp sub-microsecond modulation, system-level uniformity, and precision circuit design. Specifically:

- **Sharp temporal interfaces:** In continuum elastic media, both the operating frequency (~ 10 kHz) and wave

speed (~ 1 km/s) are high. Our analysis shows that achieving an effective temporal interface requires stiffness changes within less than $1/100$ of the wave period—i.e., sub-microsecond, spatially uniform modulation. For ~ 10 kHz flexural waves, this is realized using analog switches with ~ 0.1 μ s switching time, synchronized across all patches. Such precision is essential to generate a sharp temporal discontinuity; any mismatch would smear the interface and blur the separation of incident, reflected, and refracted waves.

- **System-level uniformity:** Achieving sharp temporal reflection in elastic media is challenging due to the need for uniform and synchronized stiffness modulation across the entire beam. We addressed this by using a centralized 180 MHz microcontroller to coordinate all analog switches with sub-microsecond precision. To eliminate spatial inhomogeneity, we identified mismatched unit cells and tuned their impedance using potentiometers until mid-beam reflections disappeared in PSV-400 measurements. With uniformity ensured, control experiments confirmed the absence of spatial reflections, and time-dependent shifts in the reflected wave verified its temporal origin, as detailed in Supplementary Section 3.
- **Component tolerance and circuit calibration:** High precision in component selection is critical. In the initial design, standard ceramic capacitors with 20% tolerance cause significant variation across piezoelectric patches, which blurs the temporal scattering response. To address this, we use high-quality film capacitors and 5%-tolerance precision resistors. This upgrade is essential for ensuring spatial uniformity and enabling clean wave splitting at the temporal interface.

We have clarified both the experimental challenge and our unique strategy in the revised manuscript. In Lines 26–28, we highlight the core difficulty as follows:

“However, realizing a sharp and spatially uniform temporal interface remains elusive in continuum elastic media, primarily due to the strict requirements of sub-microsecond synchronization and uniform stiffness modulation across the entire structure.”

We also emphasize our technical approach that enabled this realization in Lines 36-38:

“In this study, we report the first experimental realization of a sharp temporal interface in an elastic beam, enabled by real-time, circuit-driven stiffness modulation at the sub-microsecond scale using piezoelectric patches.”

Details of the experimental design and synchronization strategy are provided in Supplementary Section 2, while the tuning process, verification of spatial uniformity, and confirmation of temporal reflection are discussed in Supplementary Section 3.

Comment 2.2: I suggest revising the title. The content following the colon indicates both theory and physical observation. However, the findings of this study begin with experimental results, followed by validation against numerical simulations, and then some confirmation concerning the theoretical aspects. Thus, it may not be appropriate to claim that the paper presents a comprehensive theory. Please consider a title without a colon that accurately reflects the key contributions of the manuscript.

Response: We thank the reviewer for the helpful suggestion to revise the manuscript title. As correctly noted, the current study is not centered around a comprehensive theoretical framework, but rather begins with experimental realization, supported by simulations and theoretical interpretation.

To better reflect the structure and primary contributions of our work, we have revised the title to: “*Experimental Realization of Temporal Refraction and Reflection in Elastic Beams*”

This revised title emphasizes the following key contributions:

- It clearly identifies the **experimental realization** of temporal refraction and reflection in a elastic platform, which is the central novelty of the study.

- The term “**Elastic Beams**” highlights the use of a fundamental structural element in engineering. While the geometry is standard, its stiffness is dynamically modulated through circuit–structure coupling, enabling temporal wave control not achievable in passive systems.

We believe this revised title more accurately represents the study’s focus and scope.

Comment 2.3: The titles of the main manuscript and the Supporting Information do not match, and the numbering of authors’ affiliations is inconsistent.

Response: We thank the reviewer for pointing this out. We have revised the title in the Supporting Information to match that of the main manuscript. Additionally, we have corrected the affiliation numbering to ensure consistency between the main text and supplementary materials.

Comment 2.4: If we define metamaterials as artificial structures that demonstrate unconventional material properties or behaviors, I would appreciate a more convincing emphasis on why the beam in this study qualifies as a “meta” beam. Simply modulating wave propagation with a PZT-attached array may not inherently be considered “meta.”

Response: We thank the reviewer for this valuable comment. We fully agree with the definition that metamaterials are artificial structures designed to exhibit unconventional material behaviors. In this study, we use the term “metabeam” to describe an engineered elastic system that exhibits dynamic properties not accessible in passive or conventional structures.

Our use of “meta” reflected a systems-level perspective grounded in effective modeling. The beam with embedded piezoelectric patches, when connected to programmable circuits, functions as a subsystem with emergent, tunable stiffness. The circuits form a designed environment that shapes the mechanical behavior of the beam in time. In this sense, the metabeam represented an *effective subsystem of interest*, whose unconventional behavior including negative stiffness in the last section of Results is achieved through environmental control—a modeling philosophy consistent with many modern interpretations of metamaterials, including time-modulated, topological, and programmable mechanical systems.

Anyway, to remove confusion, we change title of the manuscript as “*Experimental Realization of Temporal Refraction and Reflection in Elastic Beams*”.

Comment 2.5: The experiments presented in the early figures were conducted at frequencies of 6–10 kHz, while the smart waveform morphing in Figure 6 selected 33 kHz. Please address the rationale behind choosing each frequency.

Response: The choice of the 6–10 kHz frequency range for observing temporal refraction and reflection is based on both physical and modeling considerations.

Although the effective bending stiffness remains positive over a broader low-frequency range, very low frequencies (e.g., below 5 kHz) result in long wavelengths that exceed the structural length of the beam. In such cases, the wave packet cannot be fully accommodated within the structure, making it difficult to observe clear wave splitting at the temporal interface. On the other hand, at higher frequencies above 10 kHz, the wavelength becomes comparable to local structural and circuit features, and minor non-idealities may begin to influence wave propagation. Therefore, the 6–10 kHz range is chosen as an optimal regime: short enough to allow multiple wave cycles within the structure for space-time analysis, yet long enough to avoid complications arising from high-frequency effects and to ensure the continuum beam model remains valid.

By contrast, the waveform morphing experiment shown in Fig. 6 is conducted at 33 kHz, which lies in the negative stiffness regime. This frequency corresponds to the critical point where the effective bending stiffness becomes zero under the circuit setting of $R_1 = 5.2 \text{ k}\Omega$. When R_1 is increased beyond 5.2 k Ω (e.g., 5.3 or 5.4 k Ω), the stiffness becomes increasingly negative at 33 kHz, and the wave attenuation becomes stronger. We exploit this tunability

to create distinct amplitude responses for waveform encoding. Therefore, the 33 kHz frequency is not arbitrarily chosen but is determined by the intersection of electrical circuit tuning and mechanical dispersion characteristics.

This design strategy allows us to selectively operate in either the propagating (positive stiffness) or evanescent (negative stiffness) regime, thereby demonstrating both temporal scattering and waveform morphing behaviors using the same experimental platform.

In addition, to clarify the rationale behind the selection of operating frequencies as requested by the reviewer, we have made the following modifications in the main text:

Line 157-158:

“The discrepancy observed for the switch from ON to OFF at 6 kHz arises because low-frequency incident signals do not terminate before reaching the temporal boundary, while the discrepancy at high frequencies is attributed to the limitations of the homogeneous beam model at short wavelengths. **Therefore, the 6–10 kHz range is deliberately selected to balance the wavelength considerations and maintain the validity of the continuum beam model.**”

Line 277-278:

“For an excitation frequency of 33 kHz, the corresponding effective bending stiffness over time is plotted in Fig. 6c and 6f. **An excitation frequency of 33 kHz is chosen because it corresponds to the transition of the effective bending stiffness from positive to negative values under circuit tuning.**”

Comment 2.6: The manuscript transitions from experimental results in the early sections, which are validated by numerical analyses, to discussing smart waveform morphing and information coding, which are initially designed and then validated through numerical results followed by experimental confirmation. Presenting the early study as yielding results that “happen to provide good results” does not create a natural narrative connecting the experimental outcomes to numerical simulations and theory. This section could benefit from a more meaningful and cohesive presentation.

Response: We thank the reviewer for the thoughtful comment. The main contribution of this manuscript is to construct a physical system to observe temporal refraction and reflection. Therefore, we present our physical construction first to demonstrate our experimental observations at the beginning, followed by the theoretical and numerical validations. After a full understanding of the temporal interfaces, we then present engineering applications of the temporal interfaces for various wave control based on the developed theoretical model, followed by experimental confirmation. We believe that the original organization of the manuscript is clear and cohesive. In fact, we present our experimental and numerical results simultaneously in Fig. 2, which naturally creates a connection between the experimental and numerical findings. The theoretical formulation further supports and interprets our results.

Reviewer 3

The authors would like to take this opportunity to thank the reviewer for their insights and suggestions. We have responded to the comments below, and the paper has been modified accordingly. All the revised and added parts have been highlighted in red in the response file, revised manuscript, and Supplementary Information.

Comment 3 (Overall): The paper addresses temporal refraction and reflection in a metabeam with time varying stiffness. In my view, there are three main contributions to the field: (A) experimental observation of temporal refraction, and corresponding Snell and Fresnel relations, for a continuous elastic medium (B) numerical simulations and theory showing potential applications in the form frequency and amplitude manipulation and (C) experiment and simulation showing an application in information coding (Morse code). The paper is well-written and clear for the most part. The SM is quite large, and one has to refer quite often to it, decreasing readability somewhat. Overall, however, the results are well explained, the analytical calculations seem to be correct and I have no concerns about the validity of the experimental data. There is a lot of interest in systems with time varying properties, and thus the paper could have a large audience in physics, material science, applied math, and more. I have a concern regarding novelty, addressed below, and a number of small suggestions to improve the manuscript. If they are addressed, the article may be suitable for publication.

Response: We sincerely thank the reviewer for the thoughtful summary and positive evaluation of our work. We fully agree with the reviewer’s identification of the three key contributions of (A), (B), and (C). In addressing the novelty of our study, we have reorganized these contributions into the following three core aspects:

(I) Physical Realization of Temporal Interfaces in Elastic Beams: Temporal interfaces—abrupt temporal changes in material properties—are extensively explored in optics, but their physical realization in elastic continua remains challenging, primarily due to the stringent requirements of sub-microsecond synchronization. Experimental demonstrations in discrete mass-spring systems (e.g., magnet-based lattices [49]) exhibit time refraction but operate at low working frequencies (~ 10 Hz), resulting in slow switching time (~ 1 ms), and cannot be extended to continuum systems due to the slow response of electromagnetic actuation. **On an experimental level**, (i) we present the first observation of a sharp temporal interface in an elastic beam via piezoelectric patches with circuit-driven modulation at high switching time ($\sim 0.1 \mu\text{s}$), enabling clear observation of temporal refraction and reflection of flexural waves; and (ii) we provide a quantitative analysis of the required transition time for property modulation to achieve ideal refraction and reflection at various operating frequencies (~ 10 kHz).

(II) Field-Theoretic Formulation of Temporal Refraction in Elasticity: Although temporal Snell’s law and Fresnel equations are well established in optics, their elastic counterparts lack a field-based formulation. **On a fundamental level**, this study advances the theory of elastodynamics by (i) deriving temporal Snell’s law and Fresnel equations for elastic beams directly from the equations of motion, and (ii) formulating momentum conservation and the breakdown of energy conservation in elastic systems with temporal interfaces using Noether’s theorem. The resulting framework is general and offers a foundation applicable to phononic, photonic, and quantum systems under temporal modulation.

(III) Engineering Waves via Temporal Interfaces: Unlike in optics, where research has advanced toward a range of wave manipulation strategies, little to no work has successfully leveraged temporal interfaces for elastic wave engineering. **On the application level**, we establish a design principle that enables simultaneous, broadband control of wave attributes—frequency, amplitude, and phase—through temporal interfaces. The resulting wave manipulation capabilities include (i) programmable frequency shifting and amplitude modulation; (ii) anti-reflection functionality via phase matching; and (iii) mechanical information encoding (e.g., Morse code). These features open new avenues for elastic signal processing, energy routing, and programmable mechanical metamaterials.

In light of these three aspects, we have revised the final two paragraphs of the revised main text to highlight the novelty and broader relevance of the study.

“Among these phenomena, wave refraction and reflection at temporal interfaces, are considered one of the most fundamental phenomena, at the foundations of time crystals, yet they are often challenging to achieve in practical wave systems. One of the key challenges is the creation of a temporal boundary, which typically requires a spatially uniform, ultrafast, and large change of wave impedance [19,48]. The associated experimental challenges keep the experimental study of wave scattering at temporal interfaces in its infancy, particularly in the context of elastic media. Recently, the observation of temporal refraction and reflection is reported for electromagnetic waves leveraging transmission-line metamaterials [19,21]. Building on this, temporal interfaces are demonstrated in discrete elastic systems using repelling magnets and electromagnetic control [49], but such platforms cannot be extended to continuum systems due to the slow response of electromagnetic actuation. In contrast, prior studies in elastic continua demonstrate smooth temporal stiffness-damping modulation for inducing amplitude modulation and spectral shaping [50], and periodic modulation for k -space bandgaps [37] and dynamic phononic crystal [51]. However, realizing a sharp and spatially uniform temporal interface remains elusive in continuum elastic media, primarily due to the strict requirements of sub-microsecond synchronization and uniform stiffness modulation across the entire structure. Although temporal Snell’s law and Fresnel equations are well established in optics and discrete elastic systems, no field-based formulation exists for continuum beams. Likewise, momentum conservation and the breakdown of energy conservation—concepts rooted in Noether’s theorem—are not formulated in time-varying beam theory. Furthermore, the use of sharp temporal interfaces for simultaneous and broadband manipulation of multiple wave attributes—frequency, amplitude, and phase—remains largely unexplored in elastic media.

Elastic beams, equipped with piezoelectric patches connected to digital and analog circuits, provide an excellent platform to achieve unconventional elastic wave phenomena, including the non-Hermitian skin effect [52], odd mass density [53], Willis responses [54], frequency conversion [55], and topological pumping [56]. In this study, we report the first experimental realization of a sharp temporal interface in an elastic beam, enabled by real-time, circuit-driven stiffness modulation at the sub-microsecond scale using piezoelectric patches. This setup allows the direct observation of temporal refraction and reflection of flexural waves. We further derive temporal Snell’s law and Fresnel equations for flexural waves directly from the governing equations of motion and formulate momentum conservation and the breakdown of energy conservation in time-varying elastic systems using Noether’s theorem. These theoretical developments are further validated through numerical simulation and experimental observation. By introducing multiple temporal interfaces, we demonstrate further control over the manipulation of flexural waves in both amplitude and frequency spectra. Finally, by programming a smooth time-varying transfer function to realize adiabatic stiffness modulations, we demonstrate additional capabilities in shaping the time-scattered waves in periodic and aperiodic fashion for smart waveform morphing and information coding. These results not only establish a programmable platform for manipulating elastic waves in practical engineering systems, but also deepen the fundamental understanding of wave-matter interactions under temporal modulation.”

Major Comments

Comment 3.1: My concern in terms of impact on the field, is that there is significant overlap in contribution (A) of their paper with the paper [48] in the bibliography. Experimental observation of temporal refraction of elastic waves, and corresponding Snell and Fresnel relations are also studied [48]. The platform is different than in the present paper, (continuous media vs discrete media). In my opinion that alone does not justify publication in Nature Communications. On the other hand, this paper also includes item (B) and (C) above, which brings the system closer to potential applications. One could also argue that the platform studied by the authors is more amenable to application (a beam vs an array of magnets). While on page 2 it is written ”a temporal interface in elastic continuum medium ... remains explored”, I believe a more detailed justification of the work’s novelty is needed. Moreover, line 27 claims that the elastic counterpart of Snell’s law and Fresnel are unexamined, yet these were explored in [48].

Response: We sincerely thank the reviewer for the thoughtful and constructive assessment. We fully understand the concern regarding the overlap with Ref. [48] (now Ref. [49] in the revised manuscript), and we appreciate the opportunity to clarify the novelty and broader contributions of our work. Below we address each aspect of the comment in turn:

- **Regarding the overlap with Ref. [49]:** Although both platforms exhibit temporal refraction, Ref. [49] does not report temporal reflection. Observing reflection requires a substantially larger stiffness change across the temporal interface, making its implementation in a continuous elastic beam far more challenging than refraction alone. Beams are ubiquitous in mechanical, civil, and aerospace structures, enabling direct integration into real-world systems. In contrast, discrete mass-spring systems (e.g., magnet-based lattices [49]) demonstrate time refraction only at low frequencies (~ 10 Hz), with slow switching times (~ 1 ms), and are limited by the slow response of electromagnetic actuation—making them unsuitable for continuum applications. Crucially, our analysis shows that achieving an ideal temporal interface requires a transition width below $1/100$ of the wave period; for ~ 10 kHz flexural waves, this demands spatially uniform stiffness modulation within sub-microsecond timescales—three orders of magnitude faster than in discrete setups.

Theoretically, while both studies derive Snell’s law and Fresnel equations, the governing equations for a continuous beam differ fundamentally from those in discrete systems. In our work, we introduce the concepts of elastic refractive index and impedance for flexural waves to express these relations in the continuum setting. Furthermore, the prior study does not address momentum conservation or the breakdown of energy conservation. While these principles have been extensively discussed in optics, their elastic counterparts remain unexplored. To fill this gap, we apply Noether’s theorem to establish momentum and energy relations in time-varying elastic systems through symmetry-based analysis.

Finally, as noted by the reviewer in contributions (B–C), unlike the discrete case that focuses solely on observing temporal refraction, our work introduces circuit-programmed temporal stiffness modulation to enable simultaneous, broadband control over multiple wave attributes—including frequency, amplitude, and phase—as well as adiabatic modulation for waveform morphing and information coding. These demonstrations highlight the engineering potential of our platform.

We have revised Lines 22-24 in the revised main text:

“Building on this, temporal interfaces are demonstrated in discrete elastic systems using repelling magnets and electromagnetic control [49], but such platforms cannot be extended to continuum systems due to the slow response of electromagnetic actuation.”

- **Regarding the novelty:** We appreciate the reviewer’s concern and agree that the original manuscript did not sufficiently justify the novelty of the contribution beyond stating that “a temporal interface in elastic continuum medium . . . remains unexplored.”

In response to Comment 3 (Overall), we have clarified and expanded our novelty claims as follows:

1) For contribution (A), in addition to the reviewer’s observation, we emphasize the significant technical challenge of realizing a sharp temporal interface in a continuous beam—specifically the need for spatially uniform, sub-microsecond stiffness modulation. This challenge distinguishes our platform from previous discrete-lattice implementations and is detailed in Novelty (I) of the response to Comment 3 (Overall).

2) For contributions (B) and (C), we follow the reviewer’s suggestion and further highlight the engineering relevance of our platform. Unlike prior work focused solely on temporal refraction, our system enables programmable frequency shifting, amplitude control, and waveform morphing, as discussed in Novelty (III).

3) Finally, in Novelty (II), we elaborate on the theoretical framework used in our study. In contrast to Ref. [49], our work not only derives temporal Snell’s law and Fresnel equations from first principles but also establishes the connection between conservation laws and symmetry using Noether’s theorem—a foundational perspective not addressed in the discrete case.

Fig. R3.1. The real part (left panel) and imaginary part (right panel) of bending stiffness with respect to the frequency.

To better highlight the novelty, we have revised the main text accordingly, as detailed at the end of our response to Comment 3 (Overall).

- **Regarding Snell’s law and Fresnel equations:** We appreciate the reviewer’s observation regarding line 27, which originally implied that the elastic counterparts of Snell’s law and Fresnel equations had not been examined. Indeed, these were explored in Ref. [49]. To clarify this point and accurately position our contribution, we have revised the statement in Lines 28–30 of the revised manuscript as follows:

“Although temporal Snell’s law and Fresnel equations are well established in optics and discrete elastic systems, no field-based formulation exists for continuum beams.”

Minor Comments

Comment 3.2: There needs to be a discussion on damping in the maintext, and why it is ignored in the model. Presumably for longer beams, as in the simulations with longer length starting on page 7), damping could play a significant role.

Response: We thank the reviewer for pointing out the problem of damping. There are two origins of damping in our system. One is from the elastic beam and another is from the circuit.

For damping from the elastic beam, we use a loss factor η to characterize the damping. By introducing the loss factor, the Young’s modulus becomes $E \rightarrow E(1 + i\eta)$ and stiffness becomes $D \rightarrow D(1 + i\eta)$. Then the dispersion relation becomes

$$\omega^2 = \frac{D(1 + i\eta)}{\rho A} k^4 \quad (26)$$

For a given frequency, the wavenumber becomes

$$k = \frac{k_0}{(1 + i\eta)^{1/4}} \quad (27)$$

where k_0 is the wavenumber without damping. For aluminum, the loss factor is 10^{-4} , therefore, the wavenumber is $k \rightarrow (1 + 2.5 \times 10^{-5}i)k_0$. Now the order of wavenumber without damping is 100 rad/m, the wavenumber with damping becomes $100 + 2.5 \times 10^{-3}i$, which means the decaying number is 2.5×10^{-3} . The wave amplitude after

propagating through our metabeam with length of 0.32 m becomes

$$A = e^{-2.5 \times 10^{-3} [\text{rad/m}] \times 0.32 [\text{m}]} A_0 = 0.9992 A_0 \quad (28)$$

where A_0 is the amplitude without damping. Therefore, the damping from elastic beam be neglected, which is consistent with our experience on experiment and simulation.

Another damping is mainly from the circuits. The resistor R_0 can introduce negative loss factor as discussed in Ref. [R3.1].

$$D = \frac{1 + 2\pi i f r R_0 [C_0 - C_p^T]}{1 + 2\pi i f r R_0 [C_0 - C_p^S]} D_0 \quad (29)$$

where $r = R_1/R_2$, $C_0 = 1.3C_p^T$, $C_p^S = C_p^T(1 - k_{31}^2)$ and the values of these parameters can be found in Table. S3. In the experiment, we tried different R_0 and find the system behaves loss effect when R_0 is big while it behaves gain effect when R_0 is small. When we choose $R_0 = 1\text{M}\Omega$, the real part and imaginary part of D/D_0 are presented in left panel and right panel of Fig. R3.1, respectively. Here, the imaginary part is negative, indicating the gain effect. The magnitude of imaginary part is the loss factor is smaller than 5×10^{-4} , so the wave amplitude after propagating through our metabeam with length of 0.32 m becomes

$$A = e^{-1.25 \times 10^{-2} [\text{rad/m}] \times 0.32 [\text{m}]} A_0 = 1.004 A_0 \quad (30)$$

where A_0 is the amplitude without damping. Therefore, the gain from circuit can be neglected.

Given that the Supplementary Materials are already quite extensive—as noted by the reviewer in the Overall Comments—and considering that damping effects are expected to be minimal over the relevant timescales, we did not expand on this point in detail, but instead included a brief discussion in Supplementary Section 11.

Note on Damping Effects: In our system, damping arises from two sources: the elastic beam and the electrical circuits. For the beam, the intrinsic damping is characterized by a low material loss factor ($\eta \sim 10^{-4}$), resulting in negligible attenuation ($< 0.1\%$ amplitude loss over the beam length). For the circuit, effective damping depends on the resistor value R_0 , which can introduce either loss or gain. By tuning $R_0 = 1 \text{ M}\Omega$, we balance these effects to minimize amplitude variation. As damping has minimal impact within the experimental timescales, it is omitted from the main theoretical model.

In Lines 127–129 of the revised manuscript, we have added the following clarification:

A detailed justification for neglecting shear deformation (via the Timoshenko beam model), as well as the negligible influence of higher-order flexural modes, longitudinal modes, damping, and nonlinear effects, is provided in Supplementary Section 11.

References

[R3.1] Jacopo Marconi, Emanuele Riva, Matteo Di Ronco, Gabriele Cazzulani, Francesco Braghin, and Massimo Ruzzene. Experimental observation of nonreciprocal band gaps in a space-time-modulated beam using a shunted piezoelectric array. *Physical Review Applied*, 13(3):031001, 2020.

Comment 3.3: Should the x axis label for fig. 2h be wavenumber rather than frequency?

Response: We thank the reviewer for catching this mistake. The reviewer is correct—the x-axis label in Fig. 2h should be wavenumber rather than frequency. We have corrected this in the revised manuscript.

Comment 3.4: When discussing amplification, the role of nonlinearity will become important, which is not addressed. Can the authors quantify/justify, based on the amplitude of the relevant signals, that ignoring the nonlinearity is justifiable, especially when application is involved?

Response: We appreciate the reviewer’s thoughtful comment. To assess the validity of neglecting nonlinearity in our analysis, we carefully consider possible sources of nonlinearity in both the mechanical structure and the electronic circuitry.

(1) *Structural nonlinearity* consists of two aspects:

(a) *Geometric nonlinearity:* This becomes significant when the transverse deflection w is no longer small compared to the beam thickness h . A widely used metric for estimating the onset of geometric nonlinearity is the slenderness-based threshold:

$$\frac{w_{\max}}{h} \sim 0.1. \quad (31)$$

In our experiments, the maximum deflection is less than 0.001 mm while the beam thickness is 2 mm, resulting in $w_{\max}/h < 5 \times 10^{-4}$, well within the linear regime.

(b) *Material nonlinearity:* The aluminum beam operates under small-strain conditions. The stress is well below the yield point, and the strain remains within the linear elastic regime. Similarly, the piezoelectric patches are driven within ± 40 V, and the electric field is below the level (around 1 kV) that induces nonlinear constitutive behavior.

(2) *Circuit nonlinearity:* Nonlinearity may arise from several components:

(a) *Passive components:* The film capacitors and resistors used in the shunting circuit have tolerances within $\pm 5\%$. Their voltage-dependent behavior is negligible in the operating voltage range (typically less than 40 V). The capacitance variation is below 1% across this range, according to manufacturer datasheets.

(b) *Operational amplifiers:* Our circuit employs the high-voltage precision amplifier OPA445, which supports supply voltages up to ± 45 V and delivers a linear output range approaching ± 40 V. In our experiments, we operate the op-amps under ± 40 V power rails, while the actual input and output signal levels remain well below ± 2.5 V. This ensures that the op-amps function fully within their linear region. We monitored their waveforms via oscilloscope during operation and confirmed that no saturation or slew-induced distortion occurred.

(3) *Experimental validation:* As shown in Figs. 2 and 3, the experimental measurements agree well with numerical simulations based on a linear model. Notably:

- No waveform distortion (e.g., harmonics, skewing, asymmetry) is observed in either the time-domain or frequency-domain signals.
- The FFT spectra remain narrowband, and the frequency peaks match those predicted by linear theory.

If significant nonlinearities were present, one would expect observable deviation in waveform shape, central frequency, or energy spread—all of which are absent.

Across both structural and circuit domains, the operating conditions remain within the linear regime. Given that the experimental results are consistent with the predictions of linear models, we conclude that the effects of nonlinearity are negligible for the signal amplitudes used in this study, and thus the linear analysis is well justified.

Similarly, although weak nonlinearity may arise under strong excitation, its effect is negligible within the operational regime of our experiments. Given that the Supplementary Materials are already extensive, we did not expand on this point in detail, but instead included a brief discussion in Supplementary Section 11

“Note on Nonlinearity: In our study, both structural and circuit-level nonlinearities are negligible under the operating conditions. The maximum beam deflection is less than 0.001 mm—well below the geometric nonlinearity threshold ($w/h < 0.1$), and material strains remain within the elastic regime. Piezoelectric patches operate below ± 40 V, avoiding nonlinear dielectric behavior. On the circuit side, capacitors and resistors exhibit minimal voltage dependence, and operational amplifiers function well within their linear range. No harmonic distortion or waveform asymmetry is observed experimentally, and all FFT spectra agree with linear theory. Thus, the system behavior is accurately captured by a

linear model, and the effect of nonlinearity can be safely ignored for the amplitude range used in this work.”

In Lines 127–129 of the revised manuscript, we have added the following

A detailed justification for neglecting shear deformation (via the Timoshenko beam model), as well as the negligible influence of higher-order flexural modes, longitudinal modes, damping, and nonlinear effects, is provided in Supplementary Section 11.

Comment 3.5: Are all simulations, with the exception of stiffness determination, performed on Eq (1)? What algorithm is used for simulation?

Response: We thank the reviewer for this question. All simulations presented in the manuscript—except for the circuit-based stiffness identification—were performed using full-structure finite element simulations in COMSOL Multiphysics, where the real beam with piezoelectric patches and time-modulated shunt impedance is modeled directly. The simulations do not solve Eq. (1) explicitly; instead, they simulate the full coupled electromechanical response of the beam. We employed the *generalized- α time integration algorithm* for numerical stability and accuracy in transient analysis.

In addition, we have clarified the time-domain simulation setup in the main text (Lines 344–345) as follows:

“The time domain analysis is conducted for the same setup as the experiment, where the transfer function is defined as a time-varying function. **The simulations are performed using the generalized- α method, with a time step set to 1/100 of the wave period to accurately resolve the dynamic response.** The boundary conditions on both sides for temporal refraction and reflection are free boundaries”

Comment 3.6: In the SM section 4A, the first sentence states “Fig 2 of main text only experiments are shown”, whereas the caption of Fig 2b is apparently simulated. Can the authors be more clear in the caption of Fig 2 of the main text what is simulation and what is experiment, and correct the text in the SM accordingly?

Response: We thank the reviewer for carefully pointing out this issue. We have revised the first sentence of Supplementary Section 4A to more accurately describe the contents of Fig. 2 in the main text. Specifically, the revised text now reads:

“**In Fig. 2(e–h), the temporal and spatial slices, along with their Fourier transforms, of the experimental spacetime diagram shown in Fig. 2(a) are presented. However, the corresponding results for the simulated spacetime diagram in Fig. 2(c) are not provided.** Here, the corresponding simulation results are shown in Fig. S6 for comparison.”

No changes were made to the main text, as the main figure captions already correctly distinguish between experimental and simulation results.

Comment 3.7: There is not much detail or explanation to where the transfer function in Fig 7 comes from. In general the three amplitudes needed for the morse coding, why 5.2, 5.4 and 5.3 for the transfer function? How sensitive is the response to the choice of these amplitudes?

Response: We thank the reviewer for this helpful question. The values 5.2 k Ω , 5.3 k Ω , and 5.4 k Ω were chosen based on a balance between functional performance and system stability:

- **5.2 k Ω** marks the point at which the effective bending stiffness begins to become negative, which leads to significant amplitude gain and thus is useful for distinguishing the “dash” signal.

- **5.4 k Ω** is the highest resistance we can apply before the system becomes unstable due to excessive feedback-induced amplification in the circuit.
- **5.3 k Ω** serves as a middle setting that produces a clearly distinguishable amplitude between the 5.2 and 5.4 k Ω cases, and is assigned to the "dot" signal in Morse coding.

Regarding sensitivity: the amplitude response is indeed sensitive in this range, which is precisely why we chose it. Small changes in resistance lead to clearly separable amplitudes, which is advantageous for signal encoding. At the same time, we verified experimentally that typical variations in resistance (due to component tolerances or thermal drift) do not lead to confusion between the logic states. Thus, the system maintains robustness while exploiting the high local sensitivity of the transfer function.

The explanation for the resistor selection is added at the end of Supplementary Section 14:

“The values 5.2 k Ω , 5.3 k Ω , and 5.4 k Ω were chosen based on a balance between functional performance and system stability:

- **5.2 k Ω** marks the point at which the effective bending stiffness begins to become negative, which leads to significant amplitude gain and thus is useful for distinguishing the "dash" signal.
- **5.4 k Ω** is the highest resistance we can apply before the system becomes unstable due to excessive feedback-induced amplification in the circuit.
- **5.3 k Ω** serves as a middle setting that produces a clearly distinguishable amplitude between the 5.2 and 5.4 k Ω cases, and is assigned to the "dot" signal in Morse coding.”

Comment 3.8: In Fig 7, the amplitudes between the dot and space and dot and dash are somewhat close to each other. What are the associated error bars with the experiment? How can it be justified that the variation between experimental runs is not greater than the difference in the amplitude of the dash,space and dot signals?

Response: We appreciate the reviewer’s careful observation. As shown in Fig. 7a, the Morse code elements—dash, space, and dot—are encoded using distinct amplitude levels of flexural waves. These levels are clearly distinguishable by eye, and the raw experimental signal is directly displayed without overlaid simulation lines to facilitate visual comparison.

We performed repeated measurements to assess experimental variability. The peak amplitude variation across different runs is within $\pm 2.5\%$. In comparison, the amplitude difference between a dash and a dot signal exceeds 30%, and that between a space and a dot is around 15%. Therefore, the signal differences are at least **6–10 times larger** than the experimental uncertainty, making the distinction between the elements robust and reliable.

For clarity, we have added a note to the Fig. 7 caption indicating this $\pm 2.5\%$ variation range:

“**Time-varying metabeam for Morse coding.** **a** Experimental demonstration of the fundamental Morse code elements—dash, space, and dot—using flexural waves with distinct amplitude levels. **The experimental variation remains within $\pm 2.5\%$; the amplitude difference between a dash and a dot exceeds 30%, while that between a space and a dot is approximately 15%.** **b** Time profile of the resistance $R_1(t)$, used to modulate the beam’s effective stiffness to generate these signals. **c** Comparison of experimental (purple) and simulated (orange) signals encoding the text “MU”.”

Response to the Reviewers' Comments

Reviewer 1

The authors would like to take this opportunity to thank the reviewer for their insights and suggestions. We have responded to the comments below, and the paper has been modified accordingly. All the revised and added parts have been highlighted in red in the response file, revised manuscript, and Supplementary Information.

Comment 1: In the revised manuscript, the authors have addressed most of the issues I raised and have significantly improved the clarity of their manuscript. However, the most critical issue still remains unresolved. The electrically shunted elastic beam used for demonstrating temporal refraction and reflection is an experimental platform that the authors have already employed in many of their previous works. While I certainly agree that using the same experimental platform can be a valuable tool for physical interpretation, if the main contribution of this study is to experimentally validate previously proposed physical phenomena (to the extent that the title claims ‘Experimental Realization’), then the experimental setup itself must also exhibit a degree of novelty.

Response: We sincerely thank the reviewer for acknowledging the improvements in the revised manuscript and for highlighting this important concern. We fully understand the reviewer’s point regarding the reuse of our electrically shunted elastic beam platform. While it is true that we have employed this platform in several prior studies, we would like to emphasize that the specific implementation in this work represents a significant departure from our previous configurations, both in technical execution and in conceptual purpose.

In particular:

1. New Capability: Sharp Temporal Interface Creation

Unlike previous works, where the stiffness modulation is either continuous, periodic, or slow, this study is the first to achieve a sharp, spatially uniform, and sub-microsecond temporal interface in a continuum elastic beam using real-time analog control. Specifically, we implement sudden transitions between electrical boundary conditions using precisely controlling the switching of shunted circuits, with sub-microsecond resolution ($0.1 \mu\text{s}$). This technical achievement directly addresses a long-standing experimental challenge in the field and enables the observation of temporal refraction and reflection—phenomena that have not been previously demonstrated in such a system.

2. Novel Experimental Functionality

The system is not merely reused—it is reengineered to enable ultrafast, broadband, and programmable stiffness modulation, which is essential for replicating a true temporal boundary condition. This new functionality enables, for the first time, experimental validation of temporal Snell’s law, Fresnel-like reflection and transmission, and the Noether-based formulation of momentum and energy evolution in time-varying elastic continua.

3. New Scientific Contribution Enabled by the Platform

While the platform shares hardware elements with previous studies, the current work leverages it to uncover previously unexplored physics—namely, sharp temporal wave scattering and broadband manipulation of multiple wave parameters (amplitude, frequency, and phase) via programmable time modulation.

We have added clarifications in the revised manuscript (paragraphs 3 and 4 of the Introduction) to more clearly distinguish this implementation from our prior work and to better highlight the novelty of the setup, both in terms of its capabilities and the phenomena it enables:

... In contrast, prior studies in elastic continua demonstrate smooth temporal stiffness-damping modulation for inducing amplitude modulation and spectral shaping [50], and periodic modulation for k -space bandgaps [37] and dynamic phononic crystal [51]. **Unlike previous works employed stiffness modulation that is continuous, periodic, or slow**, realizing a sharp and spatially uniform temporal interface

remains elusive in continuum elastic media, primarily due to the strict requirements of sub-microsecond synchronization and uniform stiffness modulation across the entire structure ...

... In this study, we report the experimental realization of a sharp temporal interface in an elastic beam, enabled by real-time, circuit-driven stiffness modulation at the sub-microsecond scale using piezoelectric patches. **Specifically, we implement sudden transitions between electrical boundary conditions by precisely controlling the switching of shunted circuits, achieving sub-microsecond resolution ($0.1 \mu s$).** This setup allows the direct observation of temporal refraction and reflection of flexural waves **to address a long-standing experimental challenge in the field.** We further derive temporal Snell's law and Fresnel equations for flexural waves directly from the governing equations of motion and formulate momentum conservation and the breakdown of energy conservation in time-varying elastic systems using Noether's theorem. These theoretical developments are further validated through numerical simulation and experimental observation, **enabled by the implementation of ultrafast, broadband, and programmable stiffness modulation.** By introducing multiple temporal interfaces, we demonstrate further control over the manipulation of flexural waves in both amplitude and frequency spectra, **which have not been explored before.** Finally, by programming a smooth time-varying transfer function to realize adiabatic stiffness modulations, we demonstrate additional capabilities in shaping the time-scattered waves in periodic and aperiodic fashion for smart waveform morphing and information coding ...